# SOLVING VIDEO INVERSE PROBLEMS USING IMAGE DIFFUSION MODELS

**Taesung Kwon**[1], **Jong Chul Ye**[2]
[1] Dept. of Bio & Brain Engineering, KAIST    [2] Kim Jae Chul Graduate School of AI, KAIST
{star.kwon, jong.ye}@kaist.ac.kr

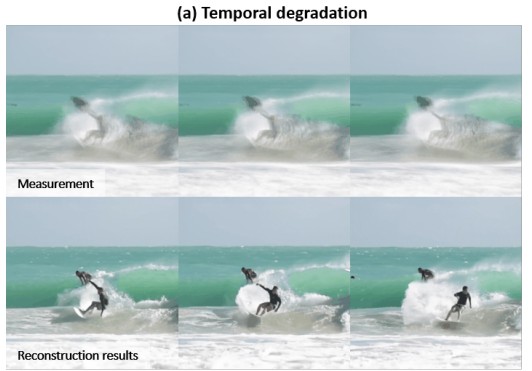
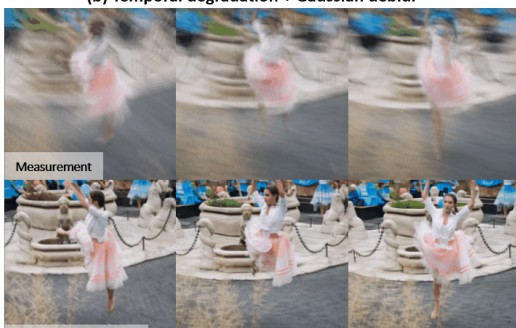
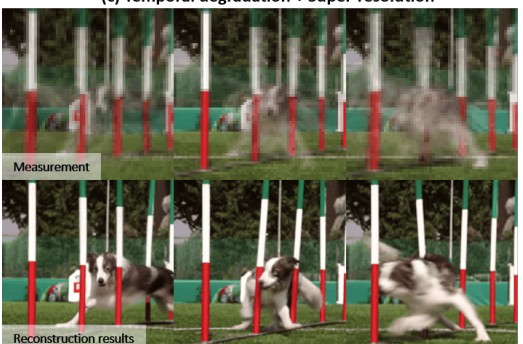
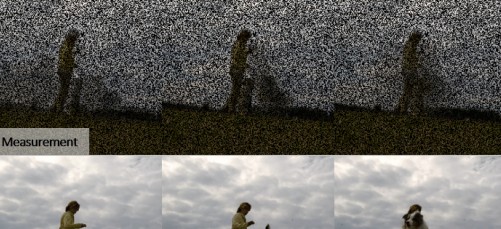

Figure 1: Representative video reconstruction results for (a) Temporal degradation, (b) Temporal degradation + Deblurring combination, (c) Temporal degradation + Super-resolution combination, and (d) Temporal degradation + Inpainting combination.

## ABSTRACT

Recently, diffusion model-based inverse problem solvers (DIS) have emerged as state-of-the-art approaches for addressing inverse problems, including image super-resolution, deblurring, inpainting, etc. However, their application to video inverse problems arising from spatio-temporal degradation remains largely unexplored due to the challenges in training video diffusion models. To address this issue, here we introduce an innovative video inverse solver that leverages only image diffusion models. Specifically, by drawing inspiration from the success of the recent decomposed diffusion sampler (DDS), our method treats the time dimension of a video as the batch dimension of image diffusion models and solves spatio-temporal optimization problems within denoised spatio-temporal batches derived from each image diffusion model. Moreover, we introduce a batch-consistent diffusion sampling strategy that encourages consistency across batches by synchronizing the stochastic noise components in image diffusion models. Our approach synergistically combines batch-consistent sampling with simultaneous optimization of denoised spatio-temporal batches at each reverse diffusion step, resulting in a novel and efficient diffusion sampling strategy for video inverse problems. Experimental results demonstrate that our method effectively addresses various spatio-temporal degradations in

video inverse problems, achieving state-of-the-art reconstructions. Project page: https://svi-diffusion.github.io/

# 1 INTRODUCTION

Diffusion models (Ho et al., 2020; Song et al., 2020) represent the state-of-the-art generative modeling by learning the underlying data distribution $p(\boldsymbol{x})$ to produce realistic and coherent data samples from the learned distribution $p_\theta(\boldsymbol{x})$. In the context of Bayesian inference, the parameterized prior distribution $p_\theta(\boldsymbol{x})$ can be disentangled from the likelihood $p(\boldsymbol{y}|\boldsymbol{x})$, which denotes the probability of observing $\boldsymbol{y}$ given $\boldsymbol{x}$. This seperation facilitates the derivation of the posterior distribution $p_\theta(\boldsymbol{x}|\boldsymbol{y}) \propto p_\theta(\boldsymbol{x})p(\boldsymbol{y}|\boldsymbol{x})$.

Diffusion model-based inverse problem solvers (DIS) (Kawar et al., 2022; Chung et al., 2022a; Song et al., 2023; Wang et al., 2023; Chung et al., 2024) leverage this property, enabling the unconditional diffusion models to solve a wide range of inverse problems. They achieve this by conditional sampling from the posterior distribution $p_\theta(\boldsymbol{x}|\boldsymbol{y})$, effectively integrating information from both the forward physics model and the measurement $\boldsymbol{y}$. This approach allows for sophisticated and precise solutions to complex inverse problems, introducing the power and flexibility of diffusion models in practical applications.

Despite extensive DIS research on a wide range of image inverse problems such as super-resolution, colorization, inpainting, compressed sensing, deblurring, and so on (Jalal et al., 2021; Kawar et al., 2022; Chung et al., 2022a; Song et al., 2023; Wang et al., 2023; Chung et al., 2024), the application of these approaches to video inverse problems, particularly those involving spatio-temporal degradation, has received relatively less attention. Specifically, in time-varying data acquisition systems, various forms of motion blur often arise due to the camera or object motions (Potmesil & Chakravarty, 1983), which can be modeled as a temporal PSF convolution of motion dynamics. These are often associated with spatial degradation caused by noise, camera defocus, and other factors. Specifically, the spatio-temporal degradation process can be formulated as:

$$\boldsymbol{Y} = \mathcal{A}(\boldsymbol{X}) + \boldsymbol{W} \tag{1}$$

with

$$\boldsymbol{X} = [\boldsymbol{x}[1] \quad \cdots \quad \boldsymbol{x}[N]], \quad \boldsymbol{Y} = [\boldsymbol{y}[1] \quad \cdots \quad \boldsymbol{y}[N]], \quad \boldsymbol{W} = [\boldsymbol{w}[1] \quad \cdots \quad \boldsymbol{w}[N]], \tag{2}$$

where $\boldsymbol{x}[n], \boldsymbol{y}[n]$ and $\boldsymbol{w}[n]$ denote the $n$-th frame ground-truth image, measurement, and additive noise, respectively; $N$ is the number of temporal frames, and $\mathcal{A}$ refers to the operator that describes the spatio-temporal degradation process. The spatio-temporal degradation introduces complexities that image diffusion priors cannot fully capture, as image diffusion priors are primarily designed to handle spatial features rather than temporal dynamics. Employing video diffusion models (Ho et al., 2022) could address these issues, but poses significant implementation challenges for video inverse problems, due to the difficulty of training video diffusion models for various applications.

Contrary to the common belief that a pre-trained video diffusion model is necessary for solving video inverse problems, here we propose a radically different method that addresses video inverse problems using only image diffusion models. Inspired by the success of the decomposed diffusion sampler (DDS) (Chung et al., 2024), which simplifies DIS by formulating it as a Krylov subspace-based optimization problem for denoised images via Tweedie's formula at each reverse sampling step, we treat the time dimension of a video as the batch dimension of image diffusion models and solve spatio-temporal optimization problems using the batch of denoised temporal frames from image diffusion models. However, treating each frame of the video as a separate sample in the batch dimension can lead to inconsistencies between temporal frames. To mitigate this, we introduce the batch-consistent sampling strategy that controls the stochastic directional component (e.g., initial noise or additive noise) of each image diffusion model during the reverse sampling process, encouraging the temporal consistency along the batch dimension. By synergistically combining batch-consistent sampling with the simultaneous optimization of the spatio-temporal denoised batch, our approach effectively addresses a range of spatio-temporal inverse problems, including spatial deblurring, super-resolution, and inpainting. Our contribution can be summarized as follows.

- We introduce an innovative video inverse problem solver using pre-trained image diffusion models by solving spatio-temporal optimization problems within the batch of denoised frames.

- We develop a batch-consistent sampling strategy to ensure temporal consistency by synchronizing stochastic noise components in image diffusion models.

- Extensive experiments confirm that our method achieves state-of-the-art performance on various video inverse problems and can also be extended to blind restoration problems.

## 2 BACKGROUND

**Diffusion models.** Diffusion models (Ho et al., 2020) attempt to model the data distribution $p_{\text{data}}(\boldsymbol{x})$ based on a latent variable model

$$p_\theta(\boldsymbol{x}_0) = \int p_\theta(\boldsymbol{x}_{0:T}) d\boldsymbol{x}_{1:T}, \quad \text{where} \quad p_\theta(\boldsymbol{x}_{0:T}) := p_\theta(\boldsymbol{x}_T) \prod_{t=1}^{T} p_\theta^{(t)}(\boldsymbol{x}_{t-1}|\boldsymbol{x}_t) \tag{3}$$

where the $\boldsymbol{x}_{1:T}$ are noisy latent variables defined by the Markov chain with Gaussian transitions

$$q(\boldsymbol{x}_t|\boldsymbol{x}_{t-1}) = \mathcal{N}(\boldsymbol{x}_t|\sqrt{\beta_t}\boldsymbol{x}_{t-1}, (1-\beta_t)I), \quad q(\boldsymbol{x}_t|\boldsymbol{x}_0) = \mathcal{N}(\boldsymbol{x}_t|\sqrt{\bar{\alpha}_t}\boldsymbol{x}_0, (1-\bar{\alpha}_t)I). \tag{4}$$

Here, the noise schedule $\beta_t$ is an increasing sequence of $t$, with $\bar{\alpha}_t := \prod_{i=1}^{t} \alpha_i$, $\alpha_i := 1 - \beta_i$. Training of diffusion models amounts to training a multi-noise level residual denoiser:

$$\min_\theta \mathbb{E}_{\boldsymbol{x}_t \sim q(\boldsymbol{x}_t|\boldsymbol{x}_0), \boldsymbol{x}_0 \sim p_{\text{data}}(\boldsymbol{x}_0), \boldsymbol{\epsilon} \sim \mathcal{N}(\boldsymbol{0}, I)} \left[ \|\boldsymbol{\epsilon}_\theta^{(t)}(\boldsymbol{x}_t) - \boldsymbol{\epsilon}\|_2^2 \right]. \tag{5}$$

Then, sampling from (3) can be implemented by ancestral sampling, which iteratively performs

$$\boldsymbol{x}_{t-1} = \frac{1}{\sqrt{\alpha_t}} \left( \boldsymbol{x}_t - \frac{1-\alpha_t}{\sqrt{1-\bar{\alpha}_t}} \boldsymbol{\epsilon}_{\theta^*}^{(t)}(\boldsymbol{x}_t) \right) + \tilde{\beta}_t \boldsymbol{\epsilon} \tag{6}$$

where $\tilde{\beta}_t := \frac{1-\bar{\alpha}_{t-1}}{1-\bar{\alpha}_t} \beta_t$ and $\theta^*$ refers to the optimized parameter from Eq. (5). On the other hand, DDIM (Song et al., 2021) accelerates the sampling based on non-Markovian assumption. Specifically, the sampling iteratively performs

$$\boldsymbol{x}_{t-1} = \sqrt{\bar{\alpha}_{t-1}} \hat{\boldsymbol{x}}_t + \sqrt{1-\bar{\alpha}_{t-1}} \hat{\boldsymbol{\epsilon}}_t \tag{7}$$

where

$$\hat{\boldsymbol{x}}_t := \frac{1}{\sqrt{\bar{\alpha}_t}} \left( \boldsymbol{x}_t - \sqrt{1-\bar{\alpha}_t} \boldsymbol{\epsilon}_{\theta^*}^{(t)}(\boldsymbol{x}_t) \right), \quad \hat{\boldsymbol{\epsilon}}_t := \frac{\sqrt{1-\bar{\alpha}_{t-1} - \eta^2 \tilde{\beta}_t^2} \boldsymbol{\epsilon}_{\theta^*}^{(t)}(\boldsymbol{x}_t) + \eta \tilde{\beta}_t \boldsymbol{\epsilon}}{\sqrt{1-\bar{\alpha}_{t-1}}} \tag{8}$$

Here, $\hat{\boldsymbol{x}}_t$ is the denoised estimate of $\boldsymbol{x}_t$ that is derived from Tweedie's formula (Efron, 2011). Accordingly, DDIM sampling can be expressed as a two-step manifold transition: (i) the noisy sample $\boldsymbol{x}_t \in \mathcal{M}_t$ transits to clean manifold $\mathcal{M}$ by deterministic estimation using Tweedie's formula, (ii) a subsequent transition from clean manifold to next noisy manifold $\mathcal{M}_{t-1}$ occurs by adding noise $\hat{\boldsymbol{\epsilon}}_t$, which is composed of the deterministic noise $\boldsymbol{\epsilon}_{\theta^*}^{(t)}(\boldsymbol{x}_t)$ and the stochastic noise $\boldsymbol{\epsilon}$.

**Diffusion model-based inverse problem solvers.** For a given loss function $\ell(\boldsymbol{x})$ which often stems from the likelihood for measurement consistency, the goal of DIS is to address the following optimization problem

$$\min_{\boldsymbol{x} \in \mathcal{M}} \ell(\boldsymbol{x}) \tag{9}$$

where $\mathcal{M}$ represents the clean data manifold sampled from unconditional distribution $p_0(\boldsymbol{x})$. Consequently, it is essential to find a way that minimizes cost while also identifying the correct manifold.

Recently, Chung et al. (2023a) proposed a general technique called diffusion posterior sampling (DPS), where the updated estimate from the noisy sample $\boldsymbol{x}_t \in \mathcal{M}_t$ is constrained to stay on the same noisy manifold $\mathcal{M}_t$. This is achieved by computing the manifold constrained gradient (MCG) (Chung et al., 2022b) on a noisy sample $\boldsymbol{x}_t \in \mathcal{M}_t$. The resulting algorithm can be stated as follows:

$$\boldsymbol{x}_{t-1} = \sqrt{\bar{\alpha}_{t-1}} \left( \hat{\boldsymbol{x}}_t - \gamma_t \nabla_{\boldsymbol{x}_t} \ell(\hat{\boldsymbol{x}}_t) \right) + \sqrt{1-\bar{\alpha}_{t-1}} \hat{\boldsymbol{\epsilon}}_t, \tag{10}$$

where $\gamma_t > 0$ denotes the step size. Under the linear manifold assumption (Chung et al., 2022b; 2023a), this allows precise transition to $\mathcal{M}_{t-1}$. Unfortunately, the computation of MCG requires computationally expensive backpropagation and is often unstable.

In a subsequent work, Chung et al. (2024) shows that under the same linear manifold assumption in DPS, the one step update by $\hat{x}_t - \gamma_t \nabla_{\hat{x}_t} \ell(\hat{x}_t)$ are guaranteed to remain within a linear subspace, thus obviating the need for explicit computation of the MCG and leading to a simpler approximation:

$$x_{t-1} \simeq \sqrt{\bar{\alpha}_{t-1}} \left( \hat{x}_t - \gamma_t \nabla_{\hat{x}_t} \ell(\hat{x}_t) \right) + \sqrt{1 - \bar{\alpha}_{t-1}} \hat{\epsilon}_t. \tag{11}$$

Furthermore, instead of using a one-step gradient update, Chung et al. (2024) demonstrated that multi-step update using Krylov subspace methods, such as the conjugate gradient (CG) method, guarantees that the intermediate steps lie in the linear subspace. This approach improves the convergence of the optimization problem without incurring additional neural function evaluations (NFE). This method, often referred to as decomposed diffusion sampling (DDS), bypasses the computation of the MCG and improves the convergence speed, making it stable and suitable for large-scale medical imaging inverse problems (Chung et al., 2024).

## 3 VIDEO INVERSE SOLVER USING IMAGE DIFFUSION MODELS

### 3.1 PROBLEM FORMULATION

Using the forward model Eq. (1) and the optimization framework in Eq. (9), the video inverse problem can be formulated as

$$\min_{X \in \mathcal{M}} \ell(X) := \|Y - \mathcal{A}(X)\|^2 \tag{12}$$

where $X$ denotes the spatio-temporal volume of the clean image composed of $N$ temporal frames as defined in Eq. (2), and $\mathcal{M}$ represents the clean video manifold sampled from unconditional distribution $p_0(X)$. Then, a naive application of the one-step gradient within the DDS framework can be formulated by

$$X_{t-1} = \sqrt{\bar{\alpha}_{t-1}} \left( \hat{X}_t - \gamma_t \nabla_{\hat{X}_t} \ell(\hat{X}_t) \right) + \sqrt{1 - \bar{\alpha}_{t-1}} \hat{\mathcal{E}}_t. \tag{13}$$

where $\hat{X}_t$ and $\hat{\mathcal{E}}_t$ refer to Tweedie's formula and noise in the spatio-temporal volume, respectively, which are defined by

$$\hat{X}_t := \frac{1}{\sqrt{\bar{\alpha}_t}} \left( X_t - \sqrt{1 - \bar{\alpha}_t} \mathcal{E}_{\theta^*}^{(t)}(X_t) \right), \quad \hat{\mathcal{E}}_t := \frac{\sqrt{1 - \bar{\alpha}_{t-1} - \eta^2 \tilde{\beta}_t^2} \mathcal{E}_{\theta^*}^{(t)}(X_t) + \eta \tilde{\beta}_t \mathcal{E}}{\sqrt{1 - \bar{\alpha}_{t-1}}} \tag{14}$$

Here, $X_t$ refers to the spatio-temporal volume at the $t$-th reverse diffusion step and $\mathcal{E} \sim \prod_{i=1}^{N} \mathcal{N}(0, I)$. Although the formula Eq. (14) is a direct extension of the image-domain counterpart Eq. (8), the main technical challenge lies in training the video diffusion model $\mathcal{E}_\theta^{(t)}$, which is required for the formula Eq. (14). Specifically, the video diffusion model is trained by

$$\min_\theta \mathbb{E}_{X_t \sim q(X_t|X_0), X_0 \sim p_{\text{data}}(X_0), \mathcal{E} \sim \prod_{i=1}^{N} \mathcal{N}(0, I)} \left[ \|\mathcal{E}_\theta^{(t)}(X_t) - \mathcal{E}\|_2^2 \right], \tag{15}$$

which requires large-scale video training data and computational resources beyond the scale of training image diffusion models. Therefore, the main research motivation is to propose an innovative method that can bypass the need for computationally extensive video diffusion models.

### 3.2 BATCH-CONSISTENT RECONSTRUCTION WITH DDS

Consider a batch of 2D diffusion models along the temporal direction:

$$\tilde{\mathcal{E}}_\theta^{(t)}(X_t) := \left[ \epsilon_{\theta^*}^{(t)}(X_t[1]) \quad \cdots \quad \epsilon_{\theta^*}^{(t)}(X_t[N]) \right] \tag{16}$$

where $\epsilon_{\theta^*}^{(t)}$ represents an image diffusion model. Suppose that $\tilde{\mathcal{E}}_\theta^{(t)}(X_t)$ is used for Eq. (14). Since unconditional reverse diffusion is entirely determined by Eq. (14), the generated video is then fully

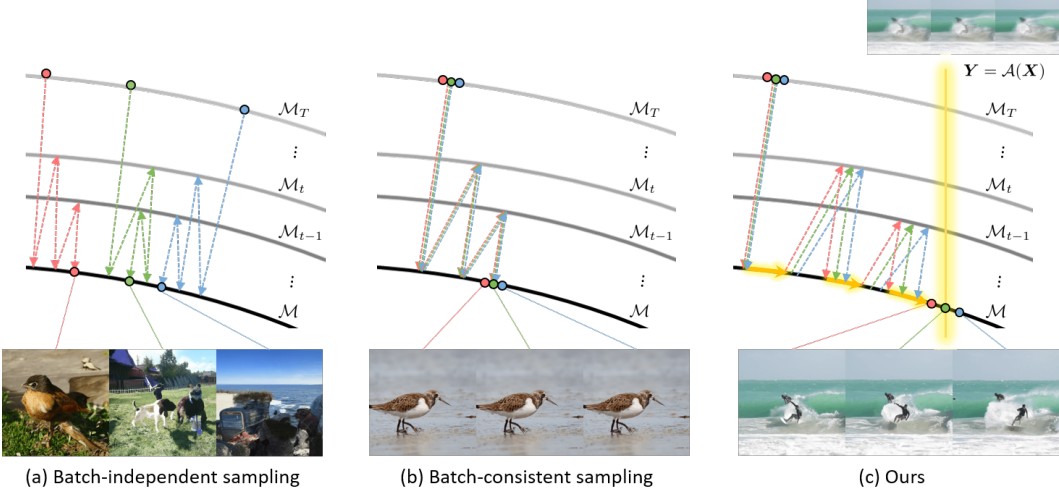

Figure 2: Geometric illustration of the sampling path evolution. (a) Batch-independent sampling produces independent frames. (b) Batch-consistent sampling produces identical frames. (c) Batch-consistent sampling combined with frame-dependent perturbation through multi-step CG generates distinct frame satisfying spatio-temporal data consistency.

controlled by the behavior of the image diffusion models. Thus, we investigate the limitations of using a batch of image diffusion models compared to using a video diffusion model and explore ways to mitigate these limitations.

Recall that for the reverse sampling of each image diffusion model, the stochastic transitions occur from two sources: (i) the initialization and (ii) re-noising. Accordingly, in batch-independent sampling, where each image diffusion model is initialized with independent random noise and re-noised with independent additive noise, it is difficult to impose any temporal consistency in video generation so that each generated temporal frame may represent different content from each other (see Fig. 2(a)). Conversely, in batch-consistent sampling, where each image diffusion model is initialized with the same noise and re-noised with the same additive noise, the generated frames from the unconditional diffusion model should be trivially reduced to identical images (see Fig. 2(b)). This dilemma is why separate video diffusion model training using Eq. (15) was considered necessary for effective video generation.

One of the most important contributions of this paper is demonstrating that the aforementioned dilemma can be readily mitigated in conditional diffusion sampling originated from inverse problems. Specifically, inspired by the DDS formulation in Eq. (13), we propose a method that employs a batch-consistent sampling scheme to ensure temporal consistency and introduces temporal diversity from the conditioning steps. More specifically, the denoised image for each frame is computed individually using Tweedie's formula via image diffusion models:

$$\hat{X}_t^b := \frac{1}{\sqrt{\bar{\alpha}_t}} \left( X_t - \sqrt{1 - \bar{\alpha}_t} \tilde{\mathcal{E}}_{\theta^*}^{(t)}(X_t) \right) \tag{17}$$

where we use the superscript $^b$ to represent the batch-consistency and $\tilde{\mathcal{E}}_{\theta^*}^{(t)}$ is a batch of image diffusion models defined by Eq. (16). Here, the image diffusion models are initialized with the same random noises to ensure temporal consistency. Subsequently, the denoised spatio-temporal batch is perturbed as a whole by applying the $l$-step conjugate gradient (CG) to optimize the data consistency term from the spatio-temporal degradation. This can be formally represented by

$$\bar{X}_t := \underset{X \in \hat{X}_t^b + \mathcal{K}_l}{\arg\min} \, \|Y - \mathcal{A}(X)\|^2 \tag{18}$$

where $\mathcal{K}_l$ denotes the $l$-dimensional Kyrlov subspace associated with the given inverse problem (Chung et al., 2024). The multistep CG can diversify each temporal frame according to the condition and achieve faster convergence than a single gradient step. The resulting solution ensures that the loss function from the spatio-temporal degradation process can be minimized with coherent but frame-by-frame distinct reconstructions. Finally, the reconstructed spatio-temporal volume from the CG is renoised with additive noise as:

$$X_{t-1} = \sqrt{\bar{\alpha}_{t-1}} \bar{X}_t + \sqrt{1 - \bar{\alpha}_{t-1}} \hat{\mathcal{E}}_t^b. \tag{19}$$

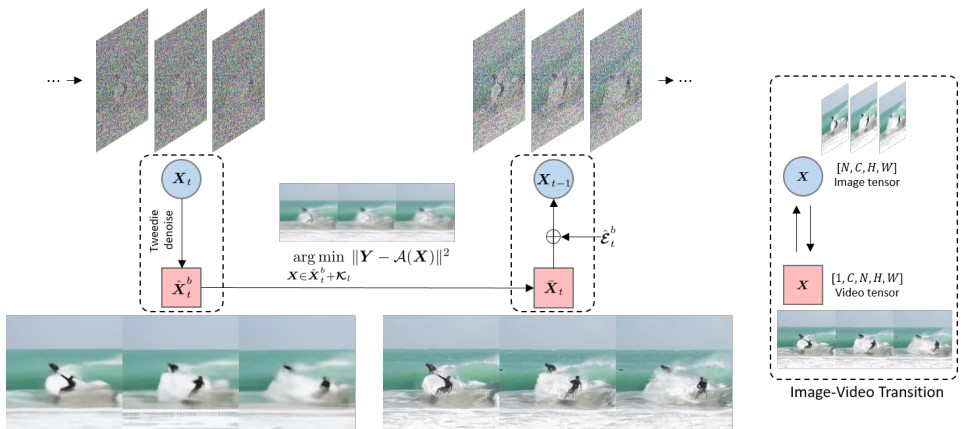

Figure 3: Sampling process in our video inverse problem solver. $\boldsymbol{X}_t$ is denoised to produce $\hat{\boldsymbol{X}}_t^b$ using 2D Tweedie formula, then reshaped into a video tensor. Multi-step CG in the video space, satisfying Eq. (18), is applied to obtain $\bar{\boldsymbol{X}}_t$, which is then reshaped back into an image batch. Finally, $\boldsymbol{X}_{t-1}$ is sampled by adding noise $\hat{\boldsymbol{\mathcal{E}}}_t^b$.

where

$$\hat{\boldsymbol{\mathcal{E}}}_t^b := \frac{\sqrt{1 - \bar{\alpha}_{t-1} - \eta^2 \tilde{\beta}_t^2} \tilde{\boldsymbol{\mathcal{E}}}_{\theta*}^{(t)}(\boldsymbol{X}_t) + \eta \tilde{\beta}_t \boldsymbol{\mathcal{E}}^b}{\sqrt{1 - \bar{\alpha}_{t-1}}} \tag{20}$$

Here, $\boldsymbol{\mathcal{E}}^b$ denotes the additive random noise from $\mathcal{N}(\boldsymbol{0}, \boldsymbol{I})$. In contrast to $\boldsymbol{\mathcal{E}}$ in Eq. (14), which is composed of frame-independent random noises, we impose batch consistency by adding the same random noises to each temporal frame to ensure temporal consistency. In summary, the proposed batch-consistent sampling and frame-dependent perturbation through multistep CG ensure that the sampling trajectory of each frame, starting from the same noise initialization, gradually diverges from each other during reverse sampling to meet the spatio-temporal data consistency. A geometric illustration of the sampling path evolution is shown in Fig. 2(c). The detailed illustration of the intermediate sampling process of our method is shown in Fig. 3. The pseudocode implementation is given in Algorithm 1. Furthermore, our method can be extended to blind inverse problems, with the corresponding algorithm and experimental results provided in Appendix B.

---

**Algorithm 1** Video inverse problem solver using 2D diffusion models

---

**Require:** $\tilde{\boldsymbol{\mathcal{E}}}_{\theta*}, T, \{\alpha_t\}_{t=1}^T, \eta, \boldsymbol{A}, \boldsymbol{Y}, l$
1: $\boldsymbol{X}_T \leftarrow \boldsymbol{\mathcal{E}}^b \sim \mathcal{N}(\boldsymbol{0}, \boldsymbol{I})$                                                             ▷ Controlled stochasticity
2: **for** $t = T : 2$ **do**
3:     $\hat{\boldsymbol{X}}_t^b \leftarrow \left(\boldsymbol{X}_t - \sqrt{1 - \bar{\alpha}_t} \tilde{\boldsymbol{\mathcal{E}}}_{\theta*}^{(t)}(\boldsymbol{X}_t)\right) / \sqrt{\bar{\alpha}_t}$                               ▷ Tweedie denoising
4:     $\bar{\boldsymbol{X}}_t \leftarrow \arg\min_{\boldsymbol{X} \in \hat{\boldsymbol{X}}_t^b + \boldsymbol{\kappa}_l} \|\boldsymbol{Y} - \mathcal{A}(\boldsymbol{X})\|^2$                   ▷ Imposing frame-dependent data consistency
5:     $\hat{\boldsymbol{\mathcal{E}}}_t^b \leftarrow \left(\sqrt{1 - \bar{\alpha}_{t-1} - \eta^2 \tilde{\beta}_t^2} \tilde{\boldsymbol{\mathcal{E}}}_{\theta*}^{(t)}(\boldsymbol{X}_t) + \eta \tilde{\beta}_t \boldsymbol{\mathcal{E}}^b\right) / \sqrt{1 - \bar{\alpha}_{t-1}}$         ▷ Controlled stochasticity
6:     $\boldsymbol{X}_{t-1} \leftarrow \sqrt{\bar{\alpha}_{t-1}} \bar{\boldsymbol{X}}_t + \sqrt{1 - \bar{\alpha}_{t-1}} \hat{\boldsymbol{\mathcal{E}}}_t^b$                                            ▷ Renoising
7: **end for**
8: $\boldsymbol{X}_0 \leftarrow (\boldsymbol{X}_1 - \sqrt{1 - \bar{\alpha}_1} \tilde{\boldsymbol{\mathcal{E}}}_{\theta*}^{(1)}(\boldsymbol{X}_1)) / \sqrt{\bar{\alpha}_1}$
9: **return** $\boldsymbol{X}_0$

---

## 4 EXPERIMENTS

In this section, we conduct thorough comparison studies to demonstrate the efficacy of the proposed method in addressing spatio-temporal degradations. Specifically, we consider two types of loss functions for video inverse problems:

$$\ell(\boldsymbol{X}) := \|\boldsymbol{Y} - \mathcal{A}(\boldsymbol{X})\|^2, \quad \ell_{TV}(\boldsymbol{X}) := \|\boldsymbol{Y} - \mathcal{A}(\boldsymbol{X})\|^2 + \lambda\, TV(\boldsymbol{X}) \tag{21}$$

where the first loss is from Eq. (12) and $TV(\boldsymbol{X})$ denotes the total variation loss along the temporal direction.

Then, classical optimization methods are used as the baselines for comparison to minimize each loss function. Specifically, the stand-alone Conjugate Gradient (CG) method is employed to minimize $\ell(\boldsymbol{X})$, while the Alternating Direction Method of Multipliers (ADMM) is used to minimize

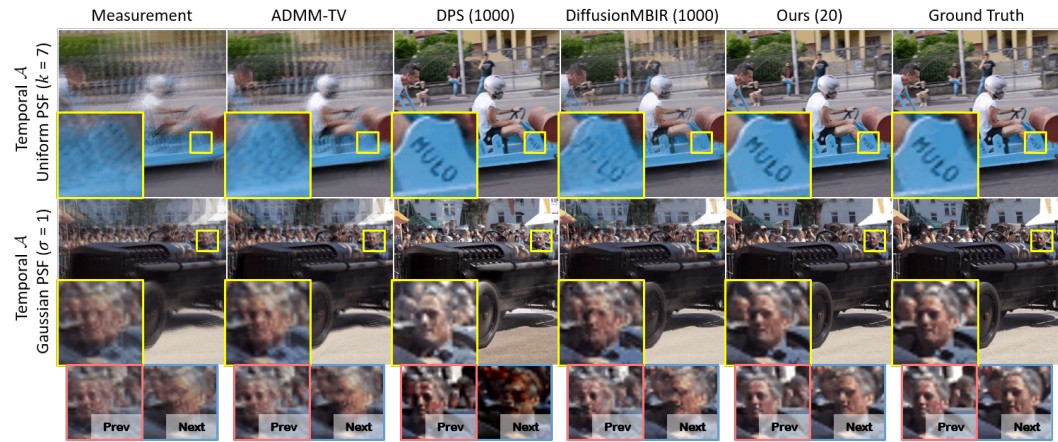

Figure 4: Qualitative evaluation of temporal degradation tasks. 1st row: temporal $\mathcal{A}$ with uniform PSF with kernel width $k = 7$. 2nd row: temporal $\mathcal{A}$ with Gaussian PSF with $\sigma=1$. Red and blue boxes indicate the enlarged views of the previous and next frames, respectively.

| Method | Time (s) | Uniform PSF ($k = 7$) | | | | Uniform PSF ($k = 13$) | | | | Gaussian PSF ($\sigma = 1.0$) | | | |
|---|---|---|---|---|---|---|---|---|---|---|---|---|---|
| | | PSNR ↑ | SSIM ↑ | LPIPS ↓ | FVD ↓ | PSNR ↑ | SSIM ↑ | LPIPS ↓ | FVD ↓ | PSNR ↑ | SSIM ↑ | LPIPS ↓ | FVD ↓ |
| Ours (20) | 12 | **43.16** | **0.992** | **0.004** | **0.008** | **39.69** | **0.984** | **0.008** | **0.035** | **34.25** | **0.959** | **0.029** | **0.068** |
| DiffusiomMBIR (1000) | 611 | 29.13 | 0.865 | 0.096 | 0.430 | 26.15 | 0.794 | 0.157 | 0.836 | 29.29 | 0.875 | 0.086 | 0.322 |
| DPS (1000) | 1244 | 33.42 | 0.914 | 0.071 | 0.325 | 20.61 | 0.666 | 0.303 | 2.055 | 12.21 | 0.610 | 0.272 | 2.210 |
| ADMM-TV | 2.4 | 24.46 | 0.744 | 0.245 | 1.297 | 23.57 | 0.697 | 0.304 | 1.580 | 26.76 | 0.826 | 0.155 | 0.656 |

Table 1: Quantitative evaluation of temporal degradation tasks on the DAVIS dataset. **Bold** indicates the best results. FVD is displayed scaled by $10^{-3}$ for easy comparison.

$\ell_{TV}(\boldsymbol{X})$. Additionally, diffusion-based methods are utilized as baselines to minimize the loss functions in Eq. (21). Specifically, DPS (Chung et al., 2022a) is used to minimize $\ell(\boldsymbol{X})$. However, instead of relying on 3D diffusion models, we use 2D image diffusion models, similar to our proposed methods, to ensure that backpropagation for MCG computation can be performed through 2D diffusion models. Second, we employ DiffusionMBIR (Chung et al., 2023b) to minimize $\ell_{TV}(\boldsymbol{X})$, also using 2D image diffusion models. Unlike the original DiffusionMBIR, which applies TV along the $z$-direction, we apply TV along the temporal direction.

To test various spatio-temporal degradations, we select the temporal degradation in time-varying data acquisition systems, which is represented as PSF convolution along temporal dimension (Potmesil & Chakravarty, 1983). We select three types of PSFs: (i) uniform PSF with widths of 7, (ii) uniform PSF with widths of 13, and (iii) Gaussian PSF with a standard deviation of 1.0. Each kernel is convolved along the temporal dimension with the ground truth video to produce the measurements. Note that convolving uniform PSF with widths of 7 and 13 correspond to averaging 7 and 13 frames, respectively. Furthermore, we combined temporal degradation and various spatial degradations to demonstrate various combinations of spatio-temporal degradations. For spatio-temporal degradations, we fix a temporal degradation as a convolving uniform PSF with a width of 7 and add various spatial degradations to the video. These spatial degradations include (i) deblurring using a Gaussian blur kernel with a standard deviation $\sigma$ of 2.0, (ii) super-resolution through a 4× average pooling, and (iii) inpainting with random masking at a ratio $r$ of 0.5 (For specific implementation details of degradations, see Appendix A).

We conduct our experiments on the DAVIS dataset (Perazzi et al., 2016; Pont-Tuset et al., 2017), which includes a wide variety of videos covering multiple scenarios. The pre-trained unconditional 256×256 image diffusion model from ADM (Dhariwal & Nichol, 2021) is used directly without fine-tuning and additional networks. All videos were normalized to the range [0, 1] and split into 16-frame samples of size 256×256. A total of 338 video samples were used for evaluation. More preprocessing details are described in the Appendix A.

For quantitative comparison, we focus on the following two widely used standard metrics: peak signal-to-noise-ratio (PSNR) and structural similarity index (SSIM) (Wang et al., 2004) with further evaluations with two perceptual metrics - Learned Perceptual Image Patch Similarity (LPIPS) (Zhang et al., 2018) and Fréchet Video Distance (FVD) (Unterthiner et al., 2019). FVD results are displayed scaled by $10^{-3}$ for easy comparison. For all proposed methods, we employ $l = 5$, $\eta = 0.15$ for 20 NFE in temporal degradation tasks, and $l = 5$, $\eta = 0.8$ for 100 NFE in spatio-temporal degradation tasks unless specified otherwise.

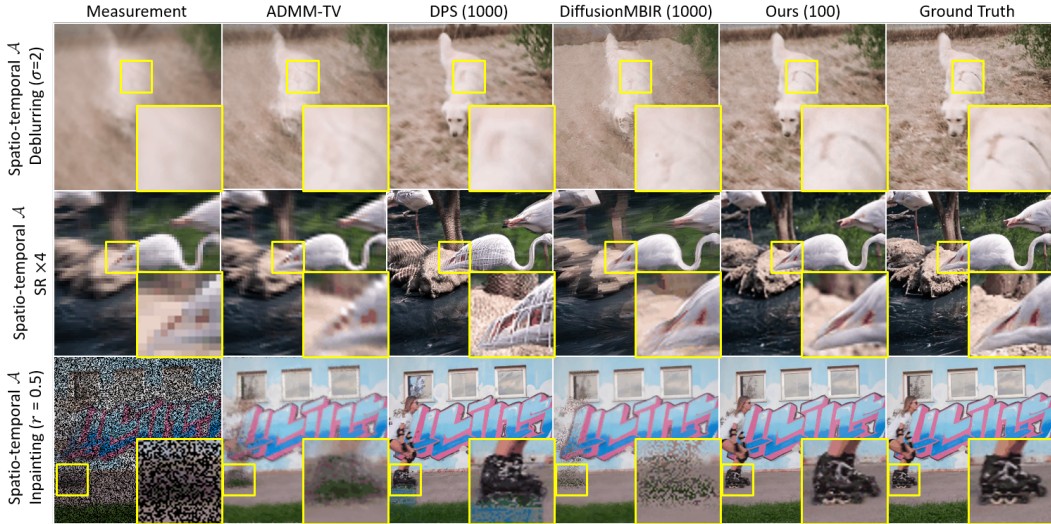

Figure 5: Qualitative evaluation of spatio-temporal degradation tasks. Each spatio-temporal degradation is combined with various spatial degradation tasks. 1$^{st}$ row: Deblurring ($\sigma = 2.0$). 2$^{nd}$ row: SR ($\times 4$). 3$^{rd}$ row: Inpainting ($r = 0.5$).

| | | + **Deblur** ($\sigma = 2.0$) | | | | + **Super-resolution** ($\times 4$) | | | | + **Inpainting** ($r = 0.5$) | | | |
| --- | --- | --- | --- | --- | --- | --- | --- | --- | --- | --- | --- | --- | --- |
| **Method** | Time (s) | PSNR ↑ | SSIM ↑ | LPIPS ↓ | FVD ↓ | PSNR ↑ | SSIM ↑ | LPIPS ↓ | FVD ↓ | PSNR ↑ | SSIM ↑ | LPIPS ↓ | FVD ↓ |
| Ours (100) | 60 | **27.77** | **0.810** | **0.270** | **0.275** | **25.71** | **0.724** | **0.279** | **0.352** | **29.45** | **0.877** | **0.047** | **0.136** |
| DiffusiomMBIR (1000) | 611 | 21.79 | 0.583 | 0.304 | 1.809 | 21.41 | 0.552 | 0.418 | 2.085 | 19.46 | 0.535 | 0.509 | 2.689 |
| DPS (1000) | 1244 | 18.19 | 0.401 | 0.602 | 3.183 | 21.39 | 0.532 | 0.318 | 1.672 | 27.43 | 0.817 | 0.115 | 0.650 |
| ADMM-TV | 2.4 | 22.76 | 0.638 | 0.462 | 1.698 | 22.09 | 0.592 | 0.469 | 1.739 | 22.53 | 0.663 | 0.326 | 1.892 |

Table 2: Quantitative evaluation of spatio-temporal degradation tasks on the DAVIS dataset. **Bold** indicates the best results. FVD is displayed scaled by $10^{-3}$ for easy comparison.

## 4.1 RESULTS

We present the quantitative results of the temporal degradation tasks in Table 1. The table shows that the proposed method outperforms the baseline methods by large margins in all metrics. The large margin improvements in FVD indicate that the proposed method successfully solves inverse problems with temporally consistent reconstruction. Fig. 4 shows the qualitative reconstruction results for temporal degradations $\mathcal{A}$. The proposed method restores much finer details compared to the baselines and demonstrates robustness across various temporal PSFs. In contrast, as shown in Fig. 4, while DPS performs well in reconstructing uniform PSFs with a kernel width of 7, it fails to accurately reconstruct frame intensities as the kernel becomes wider or more complex as shown in the bottom figures, leading to significant drops in Table 1. DiffusionMBIR ensures temporal consistency and performs well for static scenes, but it struggles with dynamic scenes in the video. In the same context, ADMM-TV produces unsatisfactory results for dynamic scenes.

The results of the spatio-temporal degradations are presented in Table 2 and Fig. 5. Even with additional spatial degradations, the proposed method consistently outperforms baseline methods. On the other hand, DPS often produces undesired details, as shown in Fig. 5. DiffusionMBIR fails to restore fine details in dynamic scenes. Specifically, in the 3$^{rd}$ row of Fig. 5, DiffusionMBIR restores the static mural painting but fails to capture the motion of the person. This is because TV regularizer often disrupts the restoration of dynamic scenes. In this context, our method ensures temporal consistency without the need for a TV regularizer. Furthermore, thanks to the consistent performance even at low NFE, the proposed method achieves a dramatic $10\times$ to $50\times$ acceleration in reconstruction time. For handling temporal degradation with 20 NFE, the proposed diffusion model-based inverse problem solver can now achieve speeds exceeding 1 FPS.

## 4.2 ABLATION STUDY

**Effect of CG updates.** Experimental results demonstrate the tangential CG updates in video space on the denoised manifold are key elements in solving spatio-temporal degradations. Here, we compare the proposed method with a stand-alone CG method to demonstrate its impact within the solver. We applied the same CG iterations as in the proposed method but excluded the diffusion updates. As shown in Fig. 6, while the stand-alone CG method nearly solves the video inverse problem, it leaves

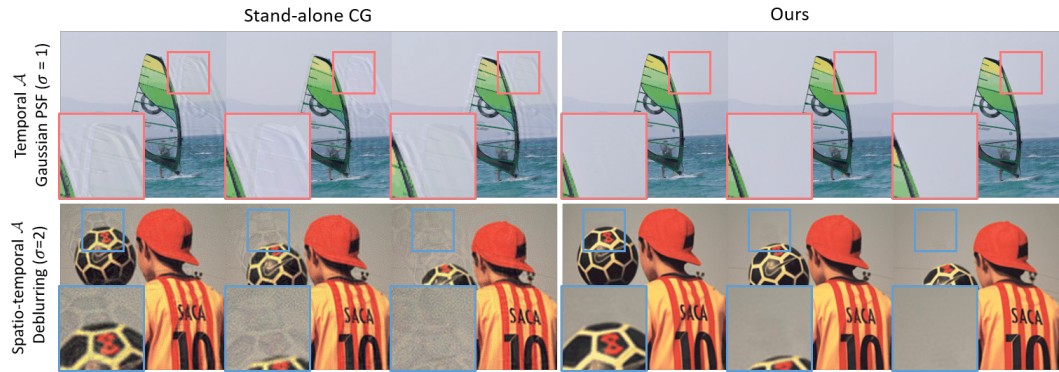

Figure 6: Reconstruction results of (left) stand-alone CG method and (right) the proposed method.

residual artifacts, as seen in the first row, or fails to fully resolve spatial degradation, as shown in the second row. In contrast, the proposed method generates natural and fully resolved frames. This indicates that the diffusion update in the proposed method refines the unnatural aspects of the CG updates.

**Effect of batch-consistent sampling.** Fig. 7 illustrates the inter-batch difference within the denoised manifold $\mathcal{M}$ during the reverse diffusion process. The blue plot shows results from our full method, while the green and orange plots represent results without stochasticity control and with gradient descent (GD) updates instead of conjugate gradient (CG) updates, respectively. Notably, GD converges more slowly than CG. Our method consistently achieves low inter-batch difference (i.e., high inter-batch similarity), ensuring batch-consistent reconstruction and precise reconstructions. In contrast, the absence of stochasticity control or the use of GD updates results in higher difference (i.e., lower similarity), leading to less consistent sampling. The intermediate samples $\hat{\boldsymbol{X}}_t$ in Fig. 7 and reconstruction results in Table 3 further confirm that our method outperforms the others in producing batch-consistent results. Further experimental results and ablation studies are illustrated in Appendix C.

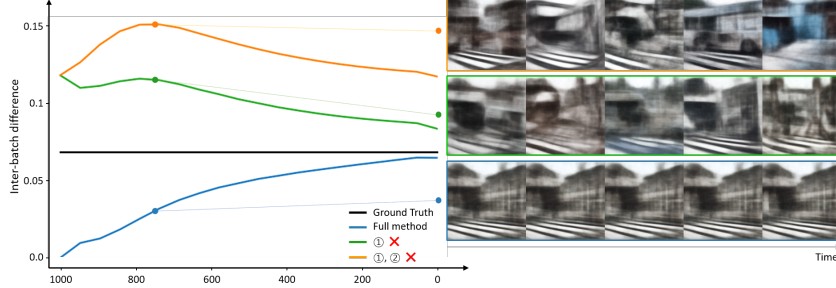

| Method | PSNR ↑ |
|---|---|
| Full | **39.69** |
| w/o ① | 30.86 |
| w/o ①, ② | 23.22 |

| Method | FVD ↓ |
|---|---|
| Full | **0.035** |
| w/o ① | 0.567 |
| w/o ①, ② | 1.275 |

Figure 7: The inter-batch difference within the denoised manifold $\mathcal{M}$, quantified as $\sum_{i=1}^{N-1} \|\hat{\boldsymbol{X}}_t[i+1] - \hat{\boldsymbol{X}}_t[i]\|/(N-1)$, throughout the reverse diffusion sampling process. ① indicates stochasticity control and ② indicates using CG updates within the denoised manifold $\mathcal{M}$.

Table 3: Reconstruction results of the ablation study.

## 5 CONCLUSION

In this work, we introduce an innovative video inverse problem solver that utilizes only image diffusion models. Our method leverages the time dimension of video as the batch dimension in image diffusion models, integrating video inverse optimization within the Tweedie denoised manifold. We combine batch-consistent sampling with video inverse optimization at each reverse diffusion step, resulting in a novel and efficient solution for video inverse problems. Extensive experiments on temporal and spatio-temporal degradations demonstrate that the proposed method achieves superior quality while being faster than previous DIS methods, even reaching speeds exceeding 1 FPS.

ACKNOWLEDGMENTS

This work was supported by the National Research Foundation of Korea under Grant RS-2024-00336454 and by the Institute of Information & Communications Technology Planning & Evaluation (IITP) grant funded by the Korea government (MSIT) (RS-2019-11190075, Artificial Intelligence Graduate School Program, KAIST).

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

# A EXPERIMENTAL DETAILS

## A.1 IMPLEMENTATION OF DEGRADATIONS

For spatio-temporal degradations, we applied temporal degradation followed by spatial degradation sequentially. We utilize spatial degradation operations for super-resolution, inpainting, and deblurring as specified in the official implementation from Wang et al. (2023) and Chung et al. (2022a). For super-resolution, we employ $4\times$ average pooling as the forward operator $\mathcal{A}$. For inpainting, we use a random mask to eliminate half of the pixels for both the forward operator $\mathcal{A}$. In deblurring, we apply a Gaussian blur with a standard deviation ($\sigma$) of 2.0 and a kernel width of 13 as the forward operator $\mathcal{A}$.

## A.2 DATA PREPROCESSING DETAILS

We conducted every experiment using train/val sets of DAVIS 2017 dataset (Perazzi et al., 2016; Pont-Tuset et al., 2017). 480p resolution dataset has a spatial resolution of 480×640. Therefore, to avoid spatial distortion, the frames were first center cropped to 480×480, then resized to a resolution of 256×256. The resizing was performed using the 'resize' function from the 'cv2' library. After that, all videos were normalized to the range [0, 1]. In the temporal dimension, the video was segmented into chunks of 16 frames starting from the first frame. Any remaining frames that did not form a complete set of 16 were dropped. Through this process, a total of 338 video samples were obtained. The detailed data preprocessing code and the preprocessed Numpy files have all been open-sourced.

## A.3 COMPARATIVE METHODS

**DiffusionMBIR (Chung et al., 2023b).** For DiffusionMBIR, we use the same pre-trained image diffusion model (Dhariwal & Nichol, 2021) with 1000 NFE sampling. The optimal $\rho$ and $\lambda$ values are obtained through grid search within the ranges [0.001, 10] and [0.0001, 1], respectively. The values are set to ($\rho$, $\lambda$) = (0.1, 0.001) for temporal degradation, and ($\rho$, $\lambda$) = (0.01, 0.01) for spatio-temporal degradation.

**DPS (Chung et al., 2022a).** For DPS, we use the same pre-trained image diffusion model (Dhariwal & Nichol, 2021) with 1000 NFE sampling. The optimal step size $\zeta$ is obtained through grid search within the range [0.01, 100]. The value is set to $\zeta = 30$ for both temporal degradation and spatio-temporal degradation. Memory issues exist when performing DPS sampling more than 5 batch sizes in NVIDIA GeForce RTX 4090 GPU with VRAM 24GB. Therefore, we divide 16-frame videos into 4-frame videos and use them for all DPS experiments.

**ADMM-TV.** Following the protocol of Chung et al. (2023b), we optimize the following objective

$$\boldsymbol{X}^* = \underset{\boldsymbol{X}}{\arg\min} \frac{1}{2}\|\mathcal{A}\boldsymbol{X} - \boldsymbol{Y}\|_2^2 + \lambda\|\boldsymbol{DX}\|_1 \tag{22}$$

where $\boldsymbol{D} = [\boldsymbol{D}_t, \boldsymbol{D}_h, \boldsymbol{D}_w]$, which corresponds to the classical TV. $t$, $h$, and $w$ represent temporal, height, and width directions, respectively. The outer iterations of ADMM are solved with 30 iterations and the inner iterations of CG are solved with 20 iterations, which are identical settings to Chung et al. (2023b). We perform a grid search to find the optical parameter values that produce the most visually pleasing solution. The parameter is set to ($\rho$, $\lambda$) = (1, 0.001). We set initial $\boldsymbol{X}$ as zeros.

# B  EXTENSION TO BLIND INVERSE PROBLEMS

## B.1  BLIND VIDEO DEBLURRING

---

**Algorithm 2** Ours (blind) - Extension to blind video deblurring

---

**Require:** $\tilde{\boldsymbol{\mathcal{E}}}_{\theta*}, T, \{\alpha_t\}_{t=1}^T, \eta, \boldsymbol{Y}, l, f_\phi$

1: $\boldsymbol{X}_{\text{pre}} \leftarrow f_\phi(\boldsymbol{Y})$ ▷ PSF estimation using pre-restoration module

2: $h_\sigma \leftarrow \arg\min_{h_\sigma} \|\boldsymbol{Y} - \boldsymbol{X}_{\text{pre}} * h_\sigma\|^2$

3: $\boldsymbol{X}_T \leftarrow \boldsymbol{\mathcal{E}}^b \sim \mathcal{N}(\boldsymbol{0}, \boldsymbol{I})$

4: **for** $t = T : 2$ **do**

5:    $\hat{\boldsymbol{X}}_t^b \leftarrow \left(\boldsymbol{X}_t - \sqrt{1 - \bar{\alpha}_t}\tilde{\boldsymbol{\mathcal{E}}}_{\theta*}^{(t)}(\boldsymbol{X}_t)\right)/\sqrt{\bar{\alpha}_t}$ ▷ Stage 1 with estimated PSF

6:    $\bar{\boldsymbol{X}}_t \leftarrow \arg\min_{\boldsymbol{X} \in \hat{\boldsymbol{X}}_t^b + \boldsymbol{\kappa}_l} \|\boldsymbol{Y} - \boldsymbol{X} * h_\sigma\|^2$

7:    $\hat{\boldsymbol{\mathcal{E}}}_t^b \leftarrow \left(\sqrt{1 - \bar{\alpha}_{t-1} - \eta^2\tilde{\beta}_t^2}\tilde{\boldsymbol{\mathcal{E}}}_{\theta*}^{(t)}(\boldsymbol{X}_t) + \eta\tilde{\beta}_t\boldsymbol{\mathcal{E}}^b\right)/\sqrt{1 - \bar{\alpha}_{t-1}}$

8:    $\boldsymbol{X}_{t-1} \leftarrow \sqrt{\bar{\alpha}_{t-1}}\bar{\boldsymbol{X}}_t + \sqrt{1 - \bar{\alpha}_{t-1}}\hat{\boldsymbol{\mathcal{E}}}_t^b$

9: **end for**

10: $\boldsymbol{X}_0 \leftarrow (\boldsymbol{X}_1 - \sqrt{1 - \bar{\alpha}_1}\tilde{\boldsymbol{\mathcal{E}}}_{\theta*}^{(1)}(\boldsymbol{X}_1))/\sqrt{\bar{\alpha}_1}$

11: $h_\sigma \leftarrow \arg\min_{h_\sigma} \|\boldsymbol{Y} - \boldsymbol{X}_0 * h_\sigma\|^2$ ▷ PSF estimation using stage 1 result

12: $\boldsymbol{X}_T \leftarrow \boldsymbol{\mathcal{E}}^b \sim \mathcal{N}(\boldsymbol{0}, \boldsymbol{I})$

13: **for** $t = T : 2$ **do**

14:    $\hat{\boldsymbol{X}}_t^b \leftarrow \left(\boldsymbol{X}_t - \sqrt{1 - \bar{\alpha}_t}\tilde{\boldsymbol{\mathcal{E}}}_{\theta*}^{(t)}(\boldsymbol{X}_t)\right)/\sqrt{\bar{\alpha}_t}$ ▷ Stage 2 with refined PSF

15:    $\bar{\boldsymbol{X}}_t \leftarrow \arg\min_{\boldsymbol{X} \in \hat{\boldsymbol{X}}_t^b + \boldsymbol{\kappa}_l} \|\boldsymbol{Y} - \boldsymbol{X} * h_\sigma\|^2$

16:    $\hat{\boldsymbol{\mathcal{E}}}_t^b \leftarrow \left(\sqrt{1 - \bar{\alpha}_{t-1} - \eta^2\tilde{\beta}_t^2}\tilde{\boldsymbol{\mathcal{E}}}_{\theta*}^{(t)}(\boldsymbol{X}_t) + \eta\tilde{\beta}_t\boldsymbol{\mathcal{E}}^b\right)/\sqrt{1 - \bar{\alpha}_{t-1}}$

17:    $\boldsymbol{X}_{t-1} \leftarrow \sqrt{\bar{\alpha}_{t-1}}\bar{\boldsymbol{X}}_t + \sqrt{1 - \bar{\alpha}_{t-1}}\hat{\boldsymbol{\mathcal{E}}}_t^b$

18: **end for**

19: $\boldsymbol{X}_0 \leftarrow (\boldsymbol{X}_1 - \sqrt{1 - \bar{\alpha}_1}\tilde{\boldsymbol{\mathcal{E}}}_{\theta*}^{(1)}(\boldsymbol{X}_1))/\sqrt{\bar{\alpha}_1}$

20: **return** $\boldsymbol{X}_0$

---

Our method can be extended to blind video inverse problems. In the standard approach to address blind deconvolution, alternating between PSF estimation and deconvolution is intuitive and effective. Since initial PSF estimation is challenging, we first use a light-weight video deblurring module, DeepDeblur (Nah et al., 2017), for pre-reconstruction and estimate the initial PSF from it. Using this PSF, we perform Stage 1 reconstruction using our method, then refine the PSF based on the resulting video. Finally, the refined PSF is used for the final (Stage 2) reconstruction. In summary, for blind deconvoltuion, our method can leverage a lightweight pre-restoration module to estimate the initial PSF and achieves the final reconstruction using the refined PSF. The detailed algorithm is given in Algorithm 2.

The GoPro dataset consists of 240 fps videos captured using a GoPro camera, with blur strengths created by averaging 7 to 13 consecutive frames (Nah et al., 2017). All experiments are conducted on the GoPro test dataset. In our method, Ours (blind) refers to blind reconstruction applied to randomly selected blur strengths between 7 and 13, while Ours (known, $k = 13$) corresponds to reconstruction at the maximum blur strength of 13 with known degradation. To highlight the effectiveness of our approach, we compared our results with the reconstructions obtained from the pre-restoration module using the GoPro test dataset (Nah et al., 2017). As shown in Fig. 8 and Table 4, our method provides further refinements, resulting in improved performance and yielding highly satisfactory results. Notably, in blind cases, inaccuracies in the initial and second PSF estimations may result in suboptimal performance compared to cases with known temporal degradation.

| Method (GoPro) | FVD ↓ | LPIPS ↓ | PSNR ↑ | SSIM ↑ |
|---|---|---|---|---|
| DeepDeblur | 0.119 | 0.116 | 30.93 | 0.904 |
| Ours (blind) | 0.058 | 0.017 | **38.98** | 0.974 |
| Ours (known, $k$=13) | **0.024** | **0.012** | 38.05 | **0.981** |

Table 4: Quantitative evaluations on video deblurring using the GoPro test dataset (Nah et al., 2017). **Bold** indicates the best and underline indicates the second best results.

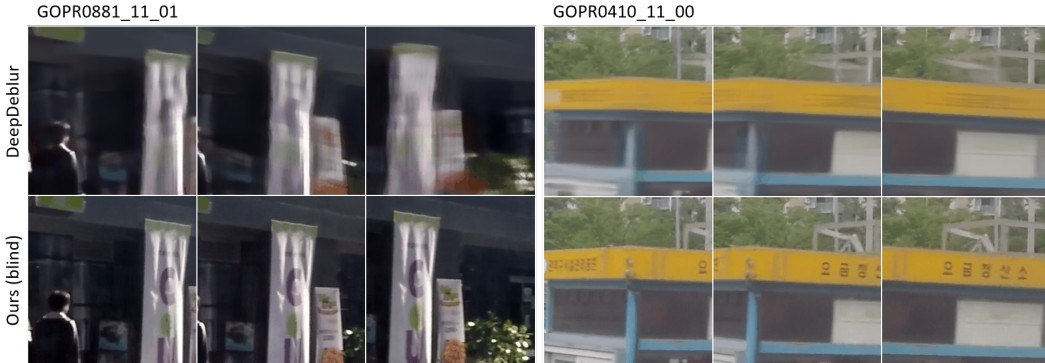

Figure 8: Reconstruction results on the GoPro test dataset (Nah et al., 2017). (Top) DeepDeblur (Nah et al., 2017) and (bottom) the proposed extension. Enlarged views of the sample are included for detailed comparison.

## B.2 BLIND VIDEO SUPER-RESOLUTION AND VIDEO FRAME INTERPOLATION

In blind video super-resolution, information about the degradation type can be inferred from the degraded measurement. For instance, if a 64×64 measurement is provided as input to a 256×256 restoration module, it is straightforward to estimate that the degradation corresponds to a 4× super-resolution (SR). Since the spatial resolution of the measurement can be directly determined, the corresponding SR process can be applied based on this estimation, enabling a simple implementation of blind video SR.

For video frame interpolation, a flow estimation module like RAFT (Teed & Deng, 2020) can be employed to generate warped estimations. Subsequently, our method can serve as an inpainting solver to fill in the gaps within the warped estimations, leveraging the explicit temporal constraints provided by batch-consistent sampling. This adaptability may allow our method to effectively tackle video interpolation tasks.

## C  FURTHER EXPERIMENTAL RESULTS AND DISCUSSIONS

### C.1  VRAM-EFFICIENT SAMPLING

The proposed method is VRAM-efficient, treating video frames as batches in the image diffusion model for sampling. As shown in Table 5, the method can reconstruct an 8-frame video at 256x256 resolution using less than 11GB of VRAM, which is feasible on GPUs like the GTX 1080Ti or RTX 2080Ti (11GB VRAM). With a single RTX 4090 GPU (24GB VRAM), it can reconstruct a 32-frame video at the same resolution.

| Frame # | VRAM (GB) |
|---------|-----------|
| 1 | 2.73 |
| 2 | 3.36 |
| 4 | 4.90 |
| 8 | 7.33 |
| 16 | 13.33 |
| 32 | 23.65 |
| RTX 4090 (24 GB) | |

Table 5: Our VRAM usage for $256 \times 256$ video.

### C.2  ABLATION STUDY OF STOCHASTICITY

Experimental results show that synchronizing stochastic noise along batch direction enables batch-consistent reconstruction, offering an effective solution for video inverse problems. While it is theoretically possible to achieve batch-consistent sampling with $\eta$ set to 0 (by eliminating stochastic noise), our empirical findings, as shown in Table 6, indicate that incorporating stochastic noise is beneficial for video reconstruction, particularly in cases involving spatio-temporal degradations. Consequently, in our experiments, the optimal $\eta$ value was determined through a grid search.

| $\eta$ | PSNR $\uparrow$ | SSIM $\uparrow$ | LPIPS $\downarrow$ | FVD $\downarrow$ |
|--------|--------|--------|--------|--------|
| 0.0 | 18.04 | 0.298 | 0.573 | 1.726 |
| 0.2 | 19.29 | 0.363 | 0.481 | 1.306 |
| 0.4 | 21.80 | 0.508 | 0.283 | 0.677 |
| 0.6 | 24.21 | 0.649 | **0.152** | 0.387 |
| 0.8 | 25.71 | 0.724 | 0.279 | **0.352** |
| 1.0 | **26.04** | **0.738** | 0.339 | 0.457 |

Table 6: Ablation study on the selection of $\eta$ for spatio-temporal degradation ($\times 4$ SR). **Bold** indicates the best and underline indicates the second best results.

### C.3  COMPARISON WITH ADDITIONAL VIDEO RESTORATION METHOD

To evaluate reconstruction performance in comparison to the latest video restoration methods, we conducted additional experiments with the recently proposed DiffIR2VR (Yeh et al., 2024), which has shown superior video processing performance over both supervised video processing method (Youk et al., 2024) and diffusion-based method SDx4 upscaler (Rombach et al., 2022). While DiffIR2VR (Yeh et al., 2024) supports video super-resolution tasks, we conducted experiments on video super-resolution ($\times 4$) task. To ensure fair comparisons with identical resolutions, we used patch reconstruction. As shown in Table 7, our method outperforms DiffIR2VR, achieving superior reconstruction performance.

| Method (DAVIS) | PSNR $\uparrow$ | FVD $\downarrow$ | LPIPS $\downarrow$ |
|----------------|--------|--------|--------|
| DiffIR2VR (Yeh et al., 2024) | 30.51 | 0.212 | **0.061** |
| Ours | **32.88** | **0.166** | 0.089 |

Table 7: Quantitative evaluations on video super-resolution ($\times 4$) using the DAVIS dataset. **Bold** indicates the best results.

## C.4 TEST ON ADDITIONAL VIDEO DATASETS

To further evaluate the adaptability of our method across diverse datasets, we conducted additional experiments on a high-frame-rate dataset (collected from Pexels[1]). For the high-frame-rate dataset from Pexels, we compared our method with DiffIR2VR (Yeh et al., 2024) on video super-resolution ($\times 4$) task. As shown in Table 8, our method maintains superior performance even on high-frame-rate data.

| Method (Pexels) | PSNR $\uparrow$ | FVD $\downarrow$ | LPIPS $\downarrow$ |
| --- | --- | --- | --- |
| DiffIR2VR (Yeh et al., 2024) | 31.31 | 0.301 | **0.056** |
| Ours | **33.79** | **0.205** | 0.104 |

Table 8: Quantitative evaluations on video super-resolution ($\times 4$) using the high frame rate (Pexels) dataset. **Bold** indicates the best results.

## C.5 HUMAN PERCEPTUAL STUDY

We conducted a perceptual human evaluation comparing our method with baseline methods used in the paper. Specifically, we collected a total of 36 votes from computer vision researchers. Reconstruction results were displayed side-by-side, and researchers were asked to vote on the method that best addressed each of the following questions: (Q1) Which video has better reconstruction quality? (Q2) Which video has better temporal consistency? As shown in Table 9, our method outperformed the baseline methods in both aspects according to human perceptual evaluations.

| Method (DAVIS) | Q1 (votes / total votes) $\uparrow$ | Q2 (votes / total votes) $\uparrow$ |
| --- | --- | --- |
| ADMM-TV | 0 | 0 |
| DPS | 0.056 | 0.056 |
| DiffusionMBIR | 0 | 0 |
| Ours | **0.944** | **0.944** |

Table 9: Human perceptual study on various video inverse problems using the DAVIS dataset. **Bold** indicates the best results.

## C.6 LIMITATIONS AND FUTURE WORKS

Our method employed the unconditional pixel-space diffusion model (Dhariwal & Nichol, 2021), which supports a maximum resolution of 256×256. Consequently, the current approach is limited by this spatial resolution. Extending the framework to latent diffusion models (Rombach et al., 2022) offers a promising direction for improving both the supported resolution and reconstruction quality, thereby enabling broader applications. In scenarios with severe temporal degradation, such as video frame interpolation, our method may become less reliable. However, as discussed in Appendix B.2, our framework is flexible enough to incorporate additional modules to address these challenges. For blind video deblurring, we utilize a two-round sampling process, which results in a doubling of the sampling time. Future research to reduce this additional sampling time could enhance the efficiency of blind inverse problem solvers.

---

[1] https://www.pexels.com/

## C.7 DETAILED VISUALIZATIONS OF EXPERIMENTAL RESULTS

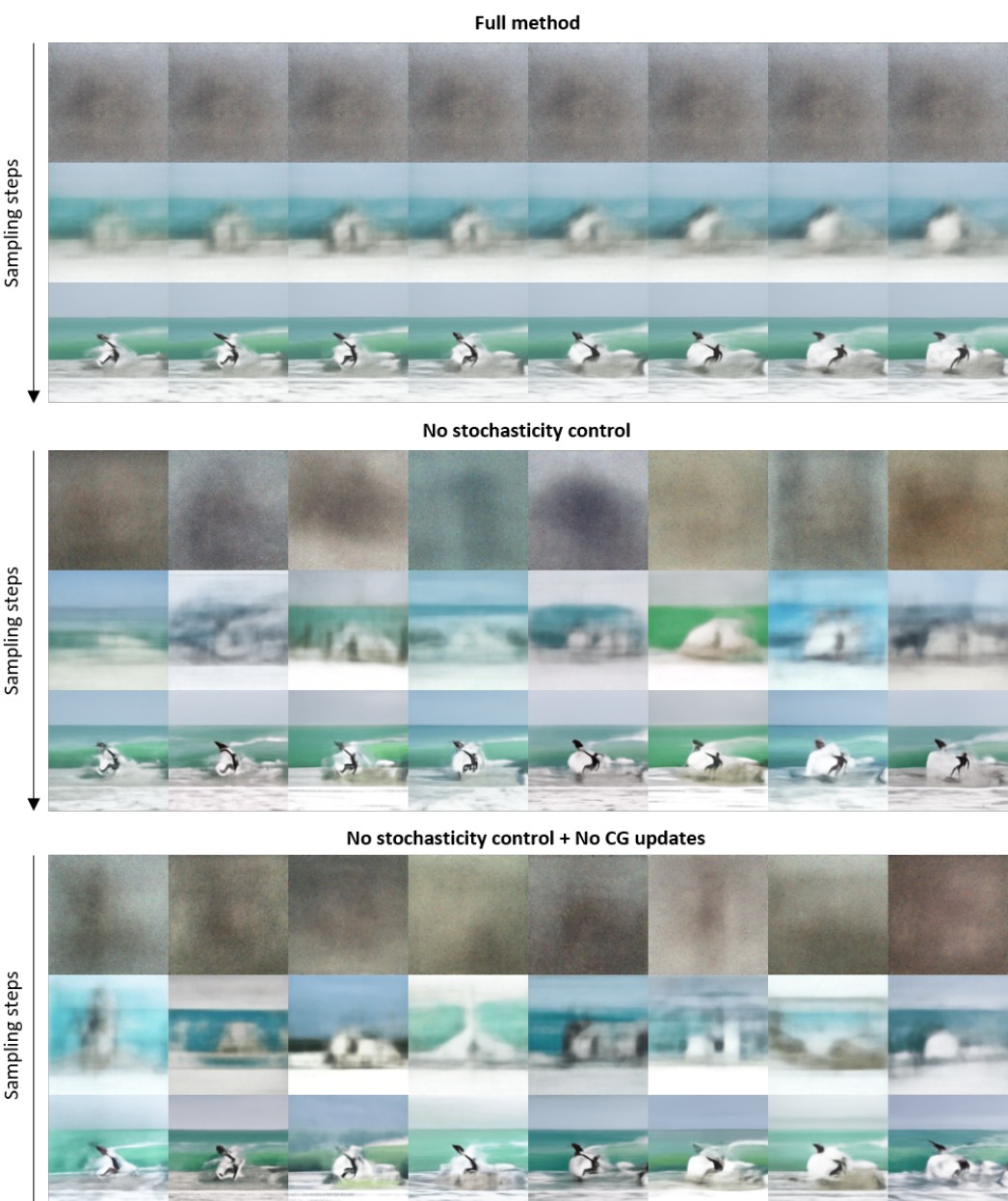

Figure 9: Ablation study results showing eight consecutive Tweedie denoised frames at different diffusion timesteps: the first row displays the 1st of 20 DDIM sampling steps, the second row displays the 5th step, and the third row displays the 10th step.

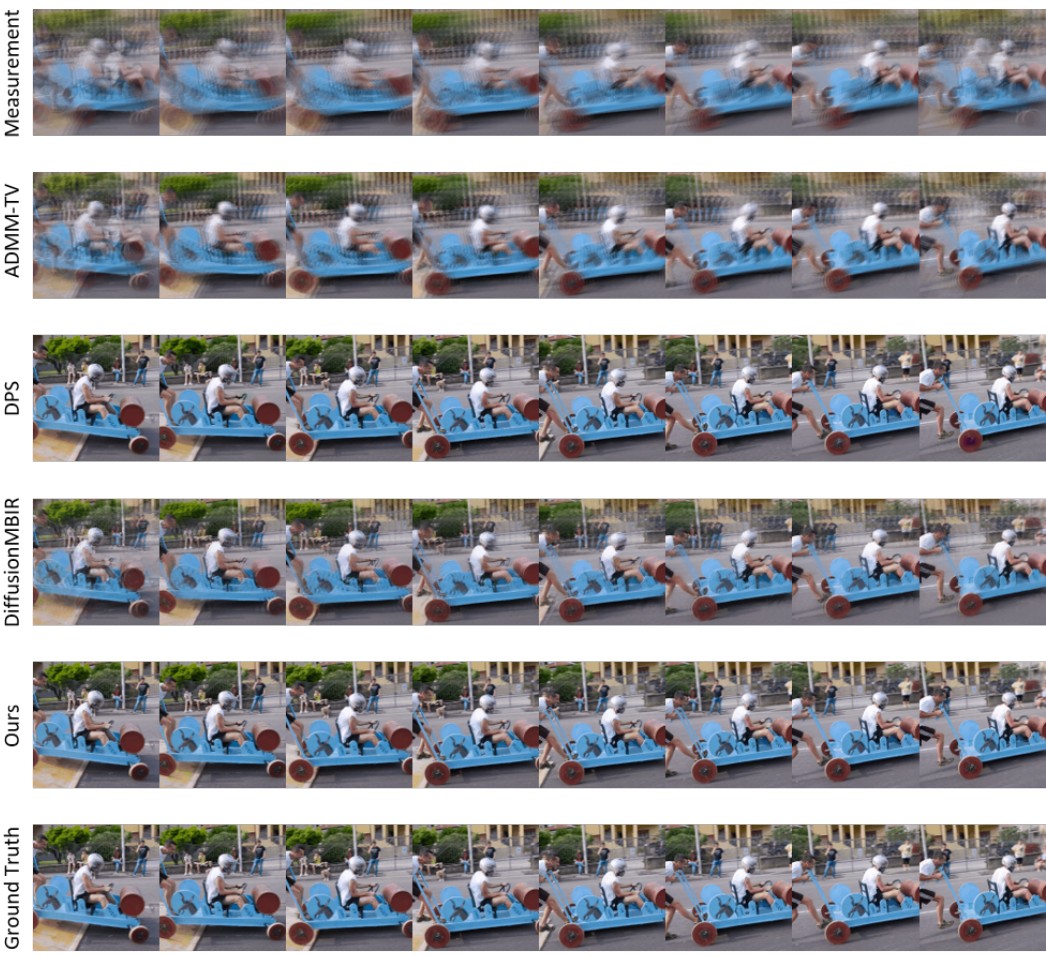

Figure 10: Detailed qualitative comparison in temporal degradation using a uniform PSF with $k$=7 on the DAVIS dataset, shown with a 2-frame skip.

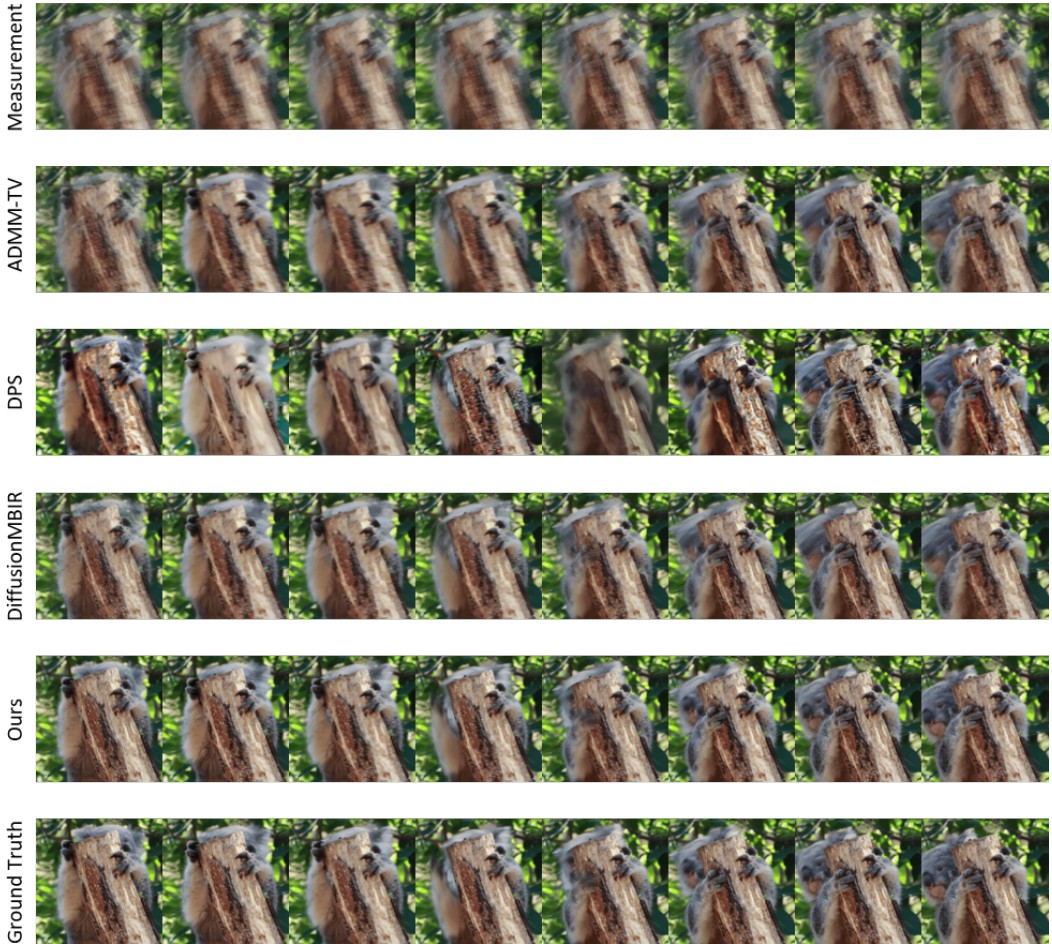

Figure 11: Detailed qualitative comparison in temporal degradation using a uniform PSF with $k$=13 on the DAVIS dataset, shown with a 2-frame skip.

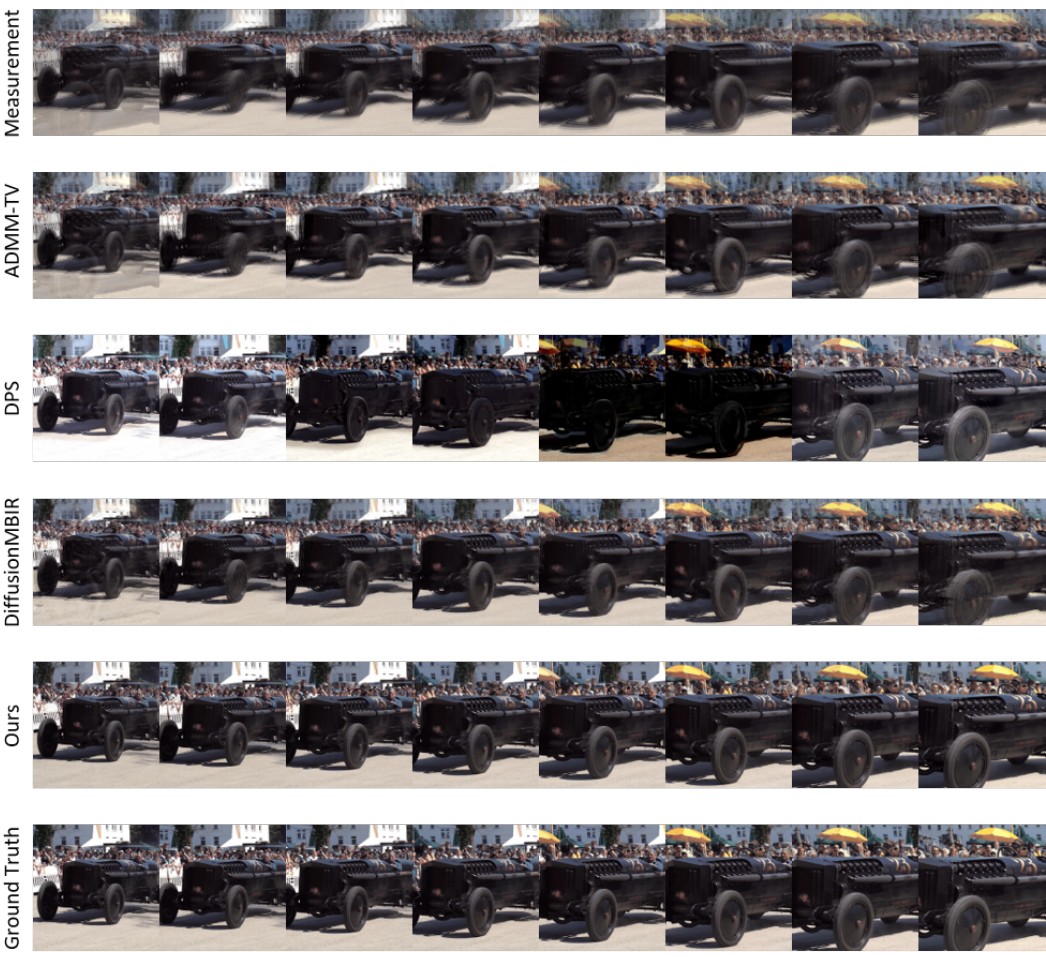

Figure 12: Detailed qualitative comparison in temporal degradation using a Gaussian PSF with $\sigma$=1 on the DAVIS dataset, shown with a 2-frame skip.

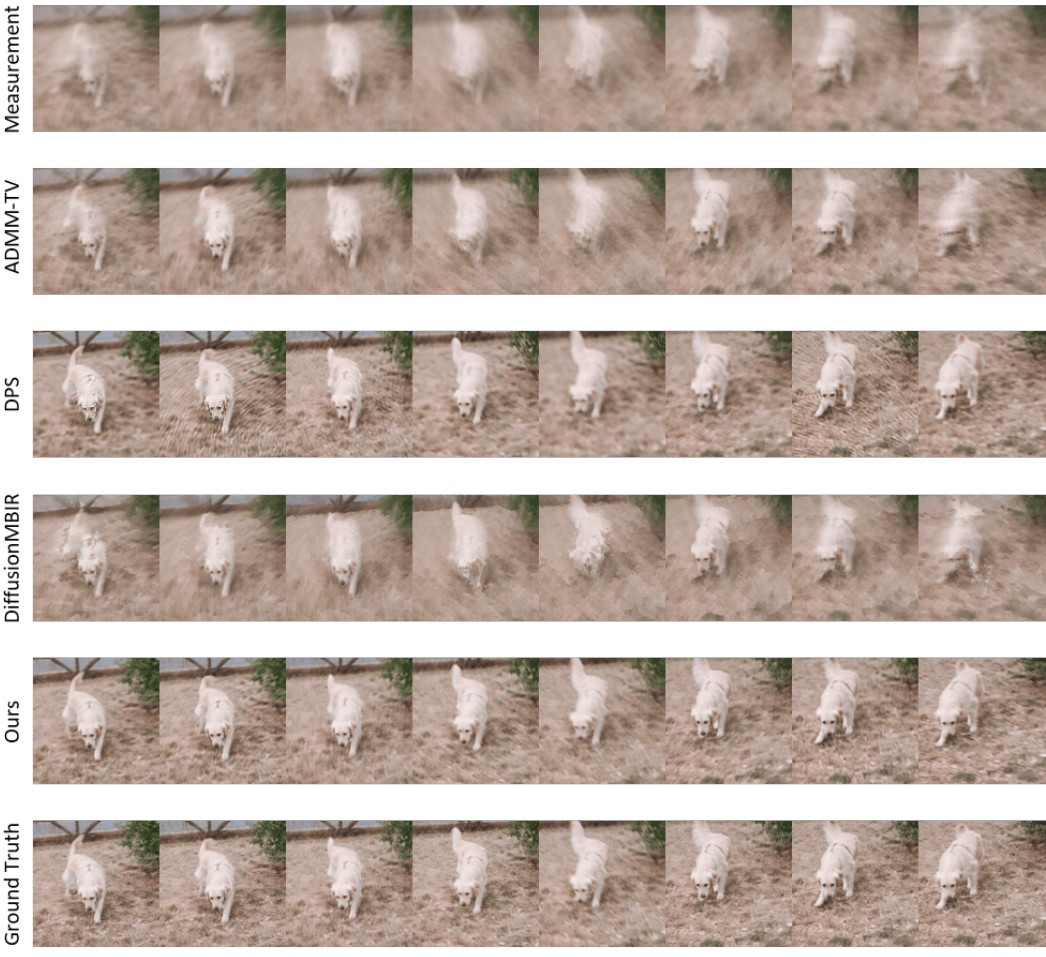

Figure 13: Detailed qualitative comparison in spatio-temporal degradation, including the spatial deblurring ($\sigma$=2.0) task on the DAVIS dataset, shown with a 2-frame skip.

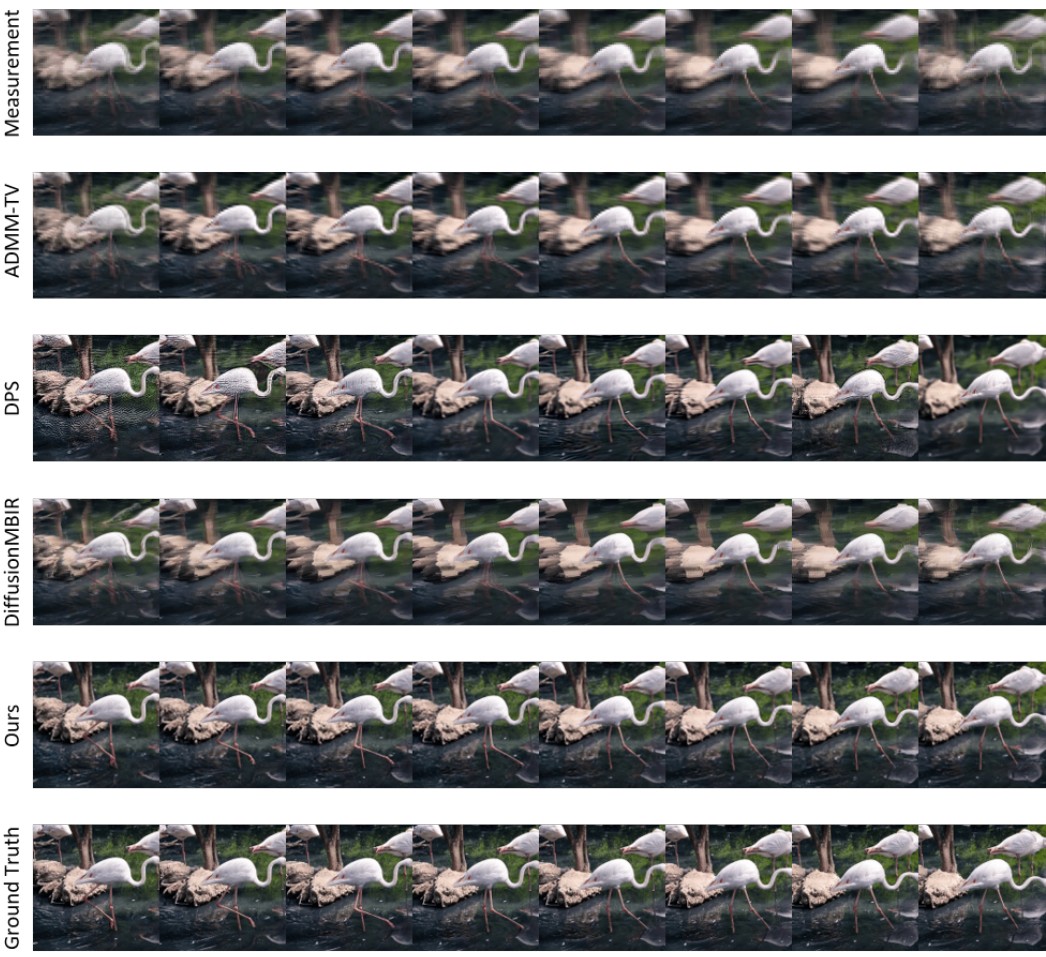

Figure 14: Detailed qualitative comparison in spatio-temporal degradation, including the spatial super-resolution ($\times$ 4) task on the DAVIS dataset, shown with a 2-frame skip.

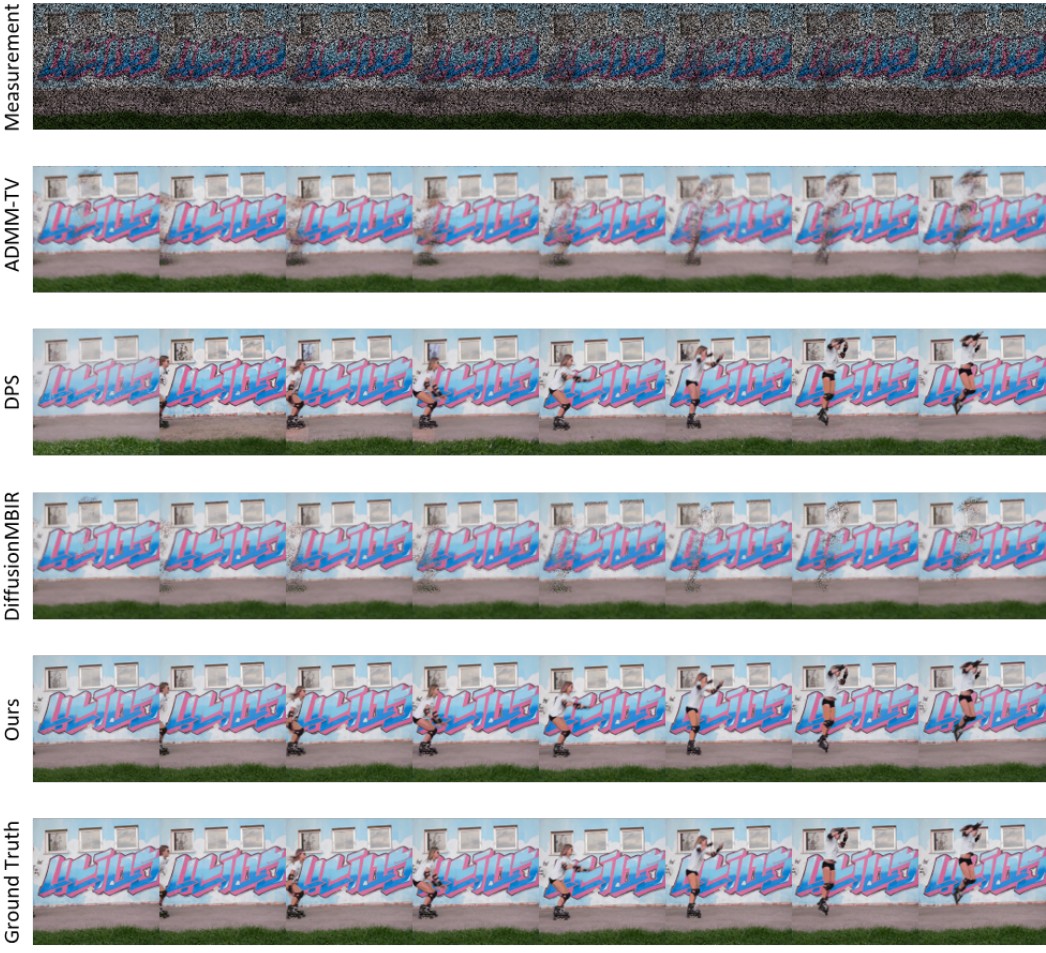

Figure 15: Detailed qualitative comparison in spatio-temporal degradation, including the spatial inpainting ($r$=0.5) task on the DAVIS dataset, shown with a 2-frame skip.

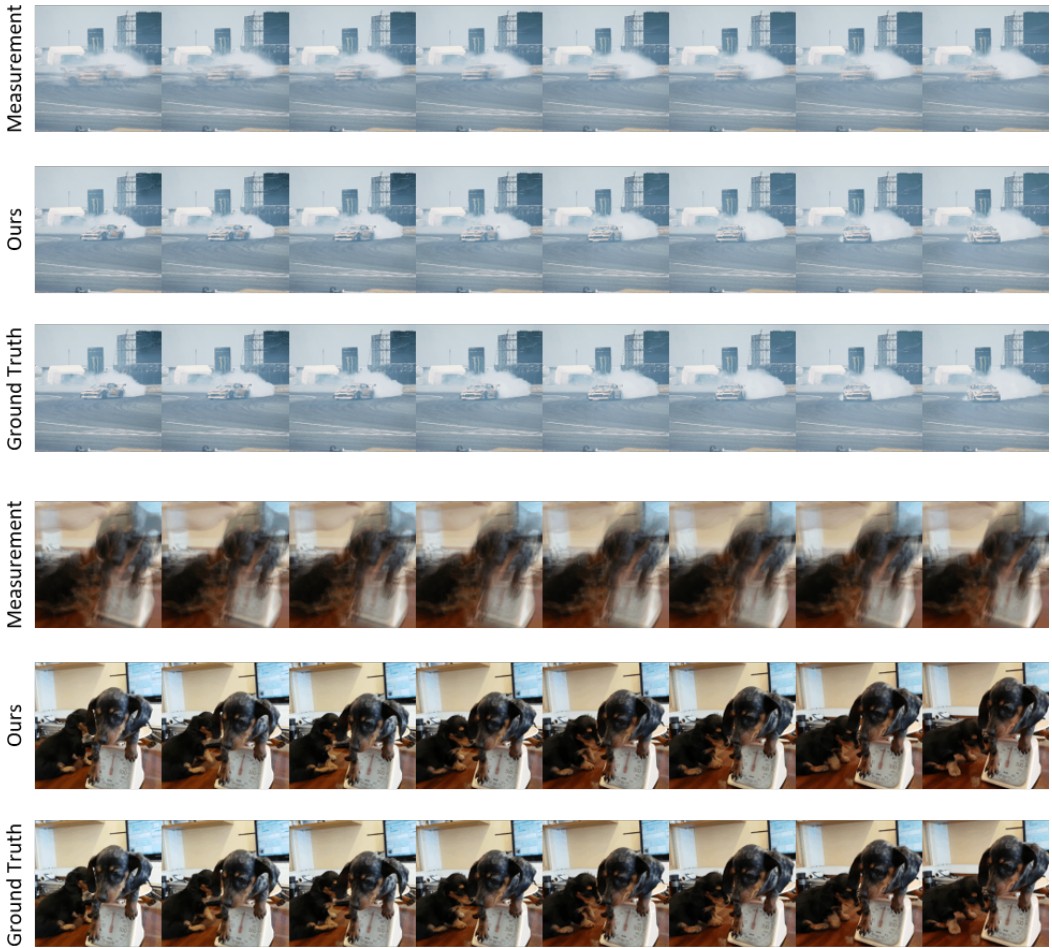

Figure 16: Additional reconstruction results for temporal degradations: (top) uniform PSF with $k$=7, (bottom) uniform PSF with $k$=13.

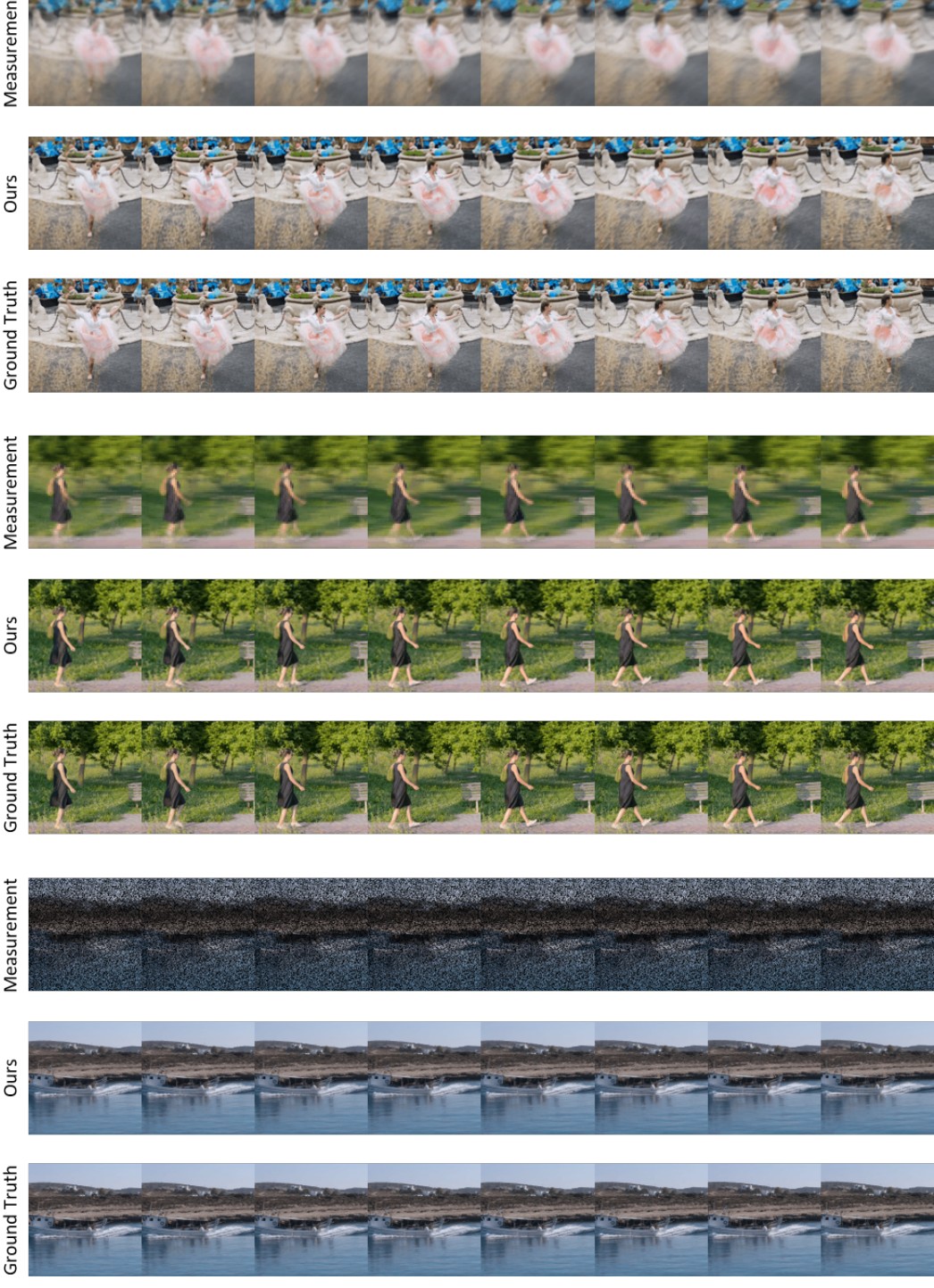

Figure 17: Additional reconstruction results for spatio-temporal degradations. The spatial degradations are: (top) deblurring ($\sigma$=2.0), (mid) super-resolution ($\times$ 4), and (bottom) inpainting ($r$=0.5).

