# OpenReview forum: "Solving Video Inverse Problems Using Image Diffusion Models"
_ICLR.cc/2025/Conference — ICLR 2025 Poster_

### Official Review · Reviewer_v7hT · 2024-10-31

**Soundness:** 3
**Presentation:** 3
**Contribution:** 2
**Rating:** 6
**Confidence:** 4

**Summary:**

The authors introduce a method to restore videos by using an pretrained image-based diffusion model.
The videos are degraded by spatio-temporal blurring, downsampling and missing data, and as far as I understand the degradation model A(X) is assumed known. The focus of the paper is on the prior on X obtained via a pre-trained diffusion model. However, in the case of videos one would need a corresponding pre-trained video diffusion model. Such models are today available, albeit very challenging to train due to the required large model, data dimensionality and compute resources. The authors suggest to use a pre-trained image diffusion model, which is much easier to train, and propose a method to adapt it to videos. The essential idea is to use the same noise used during the diffusion steps across all frames so that their denoising is more consistent. The authors show that this simple change combined with the gradients to match the data term (specifically, the conjugate gradients -- which are inspired by prior work) is sufficient to obtain temporally consistent restored videos.

**Strengths:**

The presentation is clear and self-contained. The proposed method is original as far as I know in the context of video restoration. The main originality is in the use of the same additional noise across all the frames at each step of the inverse diffusion process. This change combined with the data term constraint (and the use of conjugate gradients from prior work) seems to be sufficient to restore videos and yield sota performance on different video restoration tasks.

**Weaknesses:**

This method has been demonstrated for the case where the degradation operator is given, ie, we are
working in the non-blind video restoration case.  If I misunderstood this point, please clarify this in the rebuttal.
I am assuming that A(X) is fully known and used to compute the conjugate gradients.
My main critique is that in practice it is very difficult to have access to such operator in the case of real videos. One could argue that having these operators is at the same level of complexity of solving the restoration task. Typically, one argues that focusing on the prior term by working in the non-blind case of the restoration task is still quite useful. However, this holds only if one can show that the prior could be
used effectively also in the blind case. My main concern is that this may not hold in the proposed method.

The proposed novelty is that one should use the same additional noise samples for all the restored video frames. I wonder if this would work also in the blind case, ie, when also one needs to simultaneously recover the degradation operator. Basically, if the proposed method would only work in the non-blind case, then I would say that this is not a very significant contribution. If instead we could know that the method would also hold in the blind case, then it would be more significant (ie, the implicit temporal constraint through the same-noise sampling would be strong enough to work in the general case).

Intuitively, my concern is that the noise sampling choice implicitly guides the frame restorations to be very similar to each other, but does not provide any hint on how the temporal dynamics should be. In fact, an image diffusion model cannot impose this type of constraint. The component that imposes some temporal constraint seems to be just the degradation operator. So if this operator is unknown/to be estimated, I wonder how well all the rest would work.

In the literature there is quite a bit of past work on dealing with unknown degradation operators (the so-called _blind_ case). These methods differ from the one proposed here, as they typically use a dataset to train a model in a supervised fashion to map a blurry/degraded video to a sharp one. The advantage is that they learn to generalise to new degradation operators at test time. It would be important to understand how this approach compares to such methods and to highlight their pros and cons.
I think it would be more important to have such comparisons than to compare to even older ADMM methods.
If needed I can provide a few references on such recent blind video restoration methods, but it is straightforward to look them up.

**Questions:**

Based on the weaknesses above I have the following questions:
1) Is your method based on complete knowledge of the degradation operator?
2) If yes to 1) could you provide insights on how well your proposed video prior would work in the blind restoration case?
3) If your proposed video prior has other applications other than in the more practical blind restoration case, could you present these applications/motivations?
4) Could you clarify how your method would compare to prior blind video restoration work and why it is a better framework?

---

> ### Author Response · Authors · 2024-11-20
> **Reply to reviewer v7hT**
>
> We sincerely thank reviewer v7hT for the encouraging comments and constructive suggestions.
>
> **Q1**. Is your method based on complete knowledge of the degradation operator?
>
> **A**. See General Comment 1. The current method utilizes a known degradation operator to address video inverse problems. Extending the idea for blind restoration scenarios would be a very interesting and valuable direction, and we have incorporated our extension to deal with the blind kind.
>
> **Q2-3, W1-3**. Could you provide insights on how well your proposed video prior would work in the blind restoration case? Is there any applications other than in the more practical blind restoration case, could you present the applications/motivations?
>
> **A**. Please refer to the General comment 1. We agree that solving blind inverse problems is the ultimate goal for all inverse problem solvers. Accordingly, we demonstrate that our proposed method can be extended to address blind video inverse problems, such as video deblurring on the GoPro dataset [1]:
>
> More specifically, for blind video deblurring, we adopt a standard approach involving alternating between PSF estimation and deconvolution, which has proven intuitive and effective. Since initial PSF estimation is challenging, we first utilize a lightweight video deblurring module, DeepDeblur [1], for pre-reconstruction and estimate the initial PSF from the pre-restored video. Using this PSF, we perform Stage 1 reconstruction using the proposed method and subsequently refine the PSF based on the resulting video. The refined PSF is then used for the final (Stage 2) reconstruction.
>
> In summary, our method leverages a lightweight pre-restoration module to estimate the initial PSF and achieves superior reconstruction through iterative PSF refinement. We conducted experiments on the GoPro test dataset, and the extended method demonstrates superior performance in the blind deblurring task.
>
> | GoPro | PSNR ↑ | SSIM ↑ |
> |----------|----------|----------|
> | DeepDeblur [1]  | 30.93  | 0.904  |
> | Ours (blind) | **38.98**  | **0.974**  |
>
> Additionally, our method can be adapted to address blind superresolution. Please refer to the revised Appendix B for the detailed explanations and experimental results.
>
> **Q4, W4**. Could you clarify how your method would compare to prior blind video restoration work and why it is a better framework?
>
> **A**. As discussed in Q2-3 and W1-3, our method can be extended to address blind video inverse problems. Furthermore, our method provides state-of-the-art video reconstruction performance for cases with known degradation. To further validate its effectiveness, we conducted additional comparisons with the recently proposed blind video restoration method, DiffIR2VR [2], on 4x video super-resolution using the DAVIS and Pexels [3] datasets. Please refer to the revised Appendix C.3-4. As shown in the table, our method consistently outperforms DiffIR2VR, achieving state-of-the-art reconstruction performance.
>
> | DAVIS | PSNR ↑ | FVD ↓ |
> |----------|----------|----------|
> | DiffIR2VR [2]  | 30.51  | 212.0  |
> | Ours | **32.88**  | **166.1**  |
>
> | Pexels [3] | PSNR ↑ | FVD ↓ |
> |----------|----------|----------|
> | DiffIR2VR [2]  | 31.31  | 301.3  |
> | Ours | **33.79**  | **205.6**  |
>
> **References**
>
> [1] Nah, Seungjun, et al. "Deep multi-scale convolutional neural network for dynamic scene deblurring." CVPR 2017.
>
> [2] Yeh, Chang-Han, et al. "DiffIR2VR-Zero: Zero-Shot Video Restoration with Diffusion-based Image Restoration Models." arXiv preprint arXiv:2407.01519 (2024).
>
> [3] Pexels.com, URL: https://www.pexels.com.

---

> > ### Comment · Reviewer_v7hT · 2024-11-25
> >
> > Thanks to the authors for addressing my questions.
> > Overall I think these additional experiments suggest that
> > the proposed method could be used as a video prior also in the blind case.
> > One last issue is that the comparison on the GoPro dataset refers to the original method,
> > which dates back to 2017. Perhaps comparing with more recent works would be more
> > significant.
> > In any case, I will increase my rating. I hope the authors will incorporate all the new results
> > in the final version and also that they will introduce the use of the video prior for the more
> > practical blind case.

---

> ### Author Response · Authors · 2024-11-25
>
> Thank you for your constructive comments and for raising the score.
> It has been our pleasure to address your concerns.
> Per your request, we will incorporate the results in the final version.

---

### Official Review · Reviewer_cLVE · 2024-11-01

**Soundness:** 3
**Presentation:** 3
**Contribution:** 3
**Rating:** 6
**Confidence:** 4

**Summary:**

This paper tackles video inverse problems by leveraging pretrained image diffusion models. The authors introduce a batch-consistent sampling scheme that maintains temporal consistency while introducing temporal diversity from the conditioning steps. Experimental results demonstrate the effectiveness of this approach in preserving quality across frames.

**Strengths:**

1. The presentation is clear and intuitive, with well-defined formulations and symbols.
2. The proposed method is efficient and effective for video inverse problems. Experimental settings are clearly explained, and the results validate the effectiveness of the sampling strategy in improving temporal coherence.
3. The ablation study is comprehensive and provides a clear understanding of the role of each component.

**Weaknesses:**

Incremental Novelty: The main contribution of this paper lies in addressing the batch consistency problem by applying a 2D pretrained diffusion model. The paper highly dependent on established techniques for inverse problems, including Tweedie denoising and multi-step conjugate gradient (CG) for frame-dependent perturbation. While the results are impressive, the technical contribution appears relatively incremental.

**Questions:**

Why is the sampling approach different between the proposed method and other methods (e.g., 20 steps for DDIM vs. 1000 steps for NFE)? Would baseline methods with 20-step DDIM show a notable performance drop?

---

> ### Author Response · Authors · 2024-11-20
> **Reply to reviewer cLVE**
>
> We sincerely thank reviewer cLVE for the encouraging comments and constructive suggestions.
>
> **W1**. The technical contribution appears relatively incremental: the paper highly dependent on established techniques for inverse problems, including Tweedie denoising and multi-step conjugate gradient (CG) for frame-dependent perturbation.
>
> **A**. In contrast to your misunderstanding, a key contribution of our work is a novel batch-consistent sampling strategy, which significantly enhances temporal consistency and effectively addresses video inverse problems. Our ablation studies clearly show that individual components, such as Tweedie denoising and multi-step CG, are insufficient to solve video inverse problems independently. The proposed batch-consistent sampling strategy is essential for synergistically integrating these components to effectively address video inverse problems. For detailed results and analysis, please refer to Figure 7 and Table 3.
>
> **Q1**. Why is the sampling approach different between the proposed method and other methods? Would baseline methods with 20-step DDIM show a notable performance drop?
>
> **A**. As reported in the ablation study by DPS [1], reducing the sampling steps from 1,000 to 20 in DDIM leads to a significant performance decrease, with approximately a 0.2 increase in LPIPS. Similarly, DiffusionMBIR [2] employs full-step (2,000-step) sampling to align with its original training conditions. To ensure a fair comparison across baseline methods using ADM [3], we also apply full-step (1,000-step) sampling for each baseline, preserving their optimal performance.
>
> **References**
>
> [1] Chung, Hyungjin, et al. "Diffusion posterior sampling for general noisy inverse problems." ICLR 2023.
>
> [2] Chung, Hyungjin, et al. "Solving 3d inverse problems using pre-trained 2d diffusion models." CVPR 2023.
>
> [3] Dhariwal, Prafulla, and Alexander Nichol. "Diffusion models beat gans on image synthesis." NeurIPS 2021.

---

> > ### Comment · Reviewer_cLVE · 2024-11-27
> >
> > The rebuttal addresses my concerns. I will maintain my current score.

---

> > > ### Author Response · Authors · 2024-11-27
> > >
> > > Thank you for your thoughtful comments. It has been our pleasure to address your concerns.

---

### Official Review · Reviewer_imTd · 2024-11-03

**Soundness:** 3
**Presentation:** 3
**Contribution:** 3
**Rating:** 6
**Confidence:** 3

**Summary:**

This paper introduces a method to solve video inverse problems using image diffusion models instead of complex video models. By treating video frames as image batches, the authors achieve temporal consistency across frames through a batch-consistent sampling strategy. They also apply a multi-step optimization to improve accuracy, allowing for efficient and coherent video reconstruction without needing specialized video model training.

**Strengths:**

1. The paper utilizes image diffusion models to reduce the need for extensive video model training, leading to faster processing times.

2. The paper proposes a batch-consistent sampling strategy that ensures coherent frame generation, enhancing the visual quality of reconstructed videos.

3. The paper conducts experiments on various video inverse problems, such as deblurring and super-resolution, with improved reconstruction accuracy.

**Weaknesses:**

1. The ablation experiments in this paper could be more comprehensive. For instance, it remains unclear whether the choice of different image pre-trained models affects the final results and whether the proposed algorithm yields consistent conclusions across various pre-trained models.

2. The paper lacks testing on more complex real-world datasets, such as videos with low bitrate compression or older films.

3. The image quality in the manuscript is low, please provide a high-quality version.

**Questions:**

Can the authors provide specific examples of failure cases where your proposed method does not perform as expected?
Additionally, it is well-known that metric evaluations, even subjective ones, may not accurately reflect people's actual viewing experiences. Can the authors provide a user study to further compare the different methods?

---

> ### Author Response · Authors · 2024-11-20
> **Reply to reviewer imTd**
>
> We sincerely thank reviewer imTd for the encouraging comments and constructive suggestions.
>
> **W1**. Ablation studies on the choice of different image pre-trained models.
>
> **A**. Please refer to the General comment 3. Extending our method to latent diffusion models as an ablation study is a promising direction, and we are actively working on this as part of our future research. In response to the reviewer’s suggestion, we have included a preview example of this ongoing work. The results confirm that our method is adaptable to various diffusion models, enabling broader applications and improved performance.
>
> **W2**. Testing on real-world video restoration datasets.
>
> **A**. Please refer to the General Comment 1. In response to the reviewer’s request, we conducted video deblurring experiments on the well-known GoPro dataset [1]. To address the corresponding blind nature of video degradation, we extended our method to handle blind video deblurring. The results demonstrated consistently superior reconstruction performance.
>
> | GoPro | PSNR ↑ | SSIM ↑ |
> |----------|----------|----------|
> | DeepDeblur [1]  | 30.93  | 0.904  |
> | Ours (blind) | **38.98**  | **0.974**  |
>
> **W3**. The image quality in the manuscript is low, please provide a high-quality version.
>
> **A**. Thank you for the careful reading. Image quality issue is now fixed.
>
> **Q1**. Can the authors provide specific examples of failure cases where your proposed method does not perform as expected?
>
> **A**. In cases of severe temporal degradation, such as in video frame interpolation, our method may become less reliable. However, it is flexible enough to integrate additional modules to address such challenges. For example, a flow estimation module like RAFT [2] can be utilized to generate warped estimations. Subsequently, our method can serve as an inpainting solver, filling gaps within the warped estimations while leveraging the explicit temporal constraints provided by batch-consistent sampling. We have detailed this limitation and potential future work in the revised Appendix C.6.
>
> **Q2**. Can the authors provide a user study to further compare the different methods?
>
> **A**. Please refer to the General Comment 2. In response to the reviewer’s request, we conducted a human perceptual evaluation comparing our method with the baseline methods presented in the paper. A total of 36 votes were collected from computer vision researchers. Reconstruction results were displayed side by side, and participants were asked to vote on the method they believed performed better for the following questions:
>
> (Q1) Which video has better reconstruction quality?
>
> (Q2) Which video has better temporal consistency?
>
> As shown in the table, our method consistently outperformed the baseline methods in the human perceptual evaluations.
>
> | DAVIS | Q1 (votes / total votes) ↑ | Q2 (votes / total votes) ↑ |
> |----------|----------|----------|
> | ADMM-TV  | 0.0  | 0.0  |
> | DPS | 0.06  | 0.06  |
> | DiffusionMBIR  | 0.0  | 0.0  |
> | Ours | **0.94**  | **0.94**  |
>
> **References**
>
> [1] Nah, Seungjun, et al. "Deep multi-scale convolutional neural network for dynamic scene deblurring." CVPR 2017.
>
> [2] Teed, Zachary, and Jia Deng. "RAFT: Recurrent all-pairs field transforms for optical flow." ECCV 2020.

---

> > ### Comment · Reviewer_imTd · 2024-11-26
> >
> > The rebuttal addresses my concerns.

---

> > > ### Author Response · Authors · 2024-11-26
> > >
> > > Thank you for your constructive comments. It has been our pleasure to address your concerns.

---

### Official Review · Reviewer_Cj3w · 2024-11-04

**Soundness:** 3
**Presentation:** 3
**Contribution:** 3
**Rating:** 8
**Confidence:** 5

**Summary:**

This paper proposes a video inverse problem solver using pre-trained image diffusion models. Specifically, the method treats the time dimension of a video as the batch dimension of image diffusion modles and solves spatial-temporal optimization problems. Additionally, a batch-consistent sampling strategy is developed to ensure temporal consistency. Experimental results demonstrate the model’s effectiveness in handling various spatio-temporal degradations with significant speed improvements over existing methods.

**Strengths:**

1. This paper offers an innovative approach to video inverse problems by leveraging pre-trained image diffusion models for video tasks, eliminating the need for computationally intensive video diffusion model training.
2. The method is computationally efficient, with VRAM savings that make it feasible for deployment in lower-resource environments.

**Weaknesses:**

1. The paper lacks comparison with recent state-of-the-art video processing methods beyond diffusion-based approaches, which would provide a more comprehensive benchmark.
2. The evaluation is limited to the DAVIS dataset, potentially restricting insights into the model’s performance on a broader range of video types and characteristics, such as high-frame-rate and highly dynamic videos.
3. The degradations used in the experiments are synthesized through point spread functions (PSF), which may not fully capture real-world scenarios. Further testing on established video restoration datasets, such as GoPro for deblurring and other widely-used datasets for denoising and decompression, would help more thoroughly assess the model’s performance across diverse degradation types.

**Questions:**

1. How does the model perform on high frame rate video datasets? Could it be adapted to conduct video frame interpolation task?

---

> ### Author Response · Authors · 2024-11-20
> **Reply to reviewer Cj3w**
>
> We sincerely thank reviewer Cj3w for the encouraging comments and constructive suggestions.
>
> **W1**. Comparison with recent state-of-the-art video processing methods would provide a more comprehensive benchmark.
>
> **A**. Please refer to the General Comment 2. We conducted additional experiments on 4x video super-resolution. While we do not perform direct comparisons with multiple baseline models, we evaluated our method against DiffIR2VR [1], a method that has demonstrated superior performance compared to several prior state-of-the-art (SOTA) reconstruction methods, including SDx4 Upscaler, FMA-Net, and DiffBIR. As shown in the table, our method outperforms DiffIR2VR, achieving superior reconstruction quality.
>
> | DAVIS | PSNR ↑ | FVD ↓ |
> |----------|----------|----------|
> | DiffIR2VR [1]  | 30.51  | 212.0  |
> | Ours | **32.88** | **166.1**  |
>
> **W2, Q1**. How does the model perform on high-frame-rate video datasets?
>
> **A**. Please refer to the General Comment 2. We conducted additional experiments on high-frame-rate (high fps) video datasets. Specifically, we collected 30 high fps videos (approximately 50~60 fps) from the Pexels dataset [2], in contrast to the DAVIS dataset, which was captured at 24 fps. For these high fps videos, our method consistently achieves state-of-the-art reconstruction performance in the 4x video super-resolution task.
>
> | Pexels | PSNR ↑ | FVD ↓ |
> |----------|----------|----------|
> | DiffIR2VR [1]  | 31.31  | 301.3  |
> | Ours | **33.79** | **205.6**  |
>
> **W3**. Further testing on established video restoration datasets would help more thoroughly assess the model’s performance.
>
> **A**.	Please refer to the General Comment 1. We agree that further testing on established video restoration datasets would strengthen our work. Accordingly, we conducted video deblurring experiments on the well-known GoPro dataset [3]. To address the corresponding blind nature of video degradation, we extended our method to handle blind video deblurring.  The results demonstrated consistently superior reconstruction performance.
>
> | GoPro | PSNR ↑ | SSIM ↑ |
> |----------|----------|----------|
> | DeepDeblur [3]  | 30.93  | 0.904  |
> | Ours (blind) | **38.98** | **0.974**  |
>
> **Q2**. Could it be adapted to conduct video frame interpolation task?
>
> **A**.	Thank you for the insightful question. Our method is flexible enough to incorporate additional modules to address severe temporal degradations, such as video interpolation. For instance, a flow estimation module like RAFT [4] can be utilized to generate warped estimations. Subsequently, our method can serve as an inpainting solver to fill in the gaps within the warped estimations while leveraging the explicit temporal constraints provided by batch-consistent sampling. This adaptability suggests that our method could effectively handle video interpolation tasks. We have detailed this potential future direction in the revised Appendix B.2.
>
> **References**
>
> [1] Yeh, Chang-Han, et al. “DiffIR2VR-Zero: Zero-Shot Video Restoration with Diffusion-based Image Restoration Models.” arXiv preprint arXiv:2407.01519 (2024).
>
> [2] Pexels.com, URL: https://www.pexels.com.
>
> [3] Nah, Seungjun, et al. “Deep multi-scale convolutional neural network for dynamic scene deblurring.” CVPR 2017.
>
> [4] Teed, Zachary, and Jia Deng. “RAFT: Recurrent all-pairs field transforms for optical flow.” ECCV 2020.

---

> > ### Comment · Reviewer_Cj3w · 2024-11-25
> >
> > Thank you for the careful explanation and the additional experiments. I have read the response and considered the feedback from other reviewers. My concerns are addressed. Generally, this work introduces an innovative approach to tackling video inverse problems, and the proposed method demonstrates both effectiveness and flexibility across various video restoration tasks. I will maintain my current score.

---

> > > ### Author Response · Authors · 2024-11-25
> > >
> > > Thank you for your encouraging comments. It has been our pleasure to address your concerns.

---

### Author Response · Authors · 2024-11-20
**General response**

We would like to thank the reviewers for their constructive and thorough reviews.

We are encouraged that the reviewers view our paper as **offering an innovative approach to video inverse problems, with computational efficiency** (Cj3w), **diverse experiments showing improved reconstruction accuracy** (imTd), **clear and intuitive presentation of an efficient and effective method** (cLVE), and **the idea is original in the context of video restoration that achieve state-of-the-art performance** (v7hT).

We have made the following major revisions to address the concerns raised from the reviewers.

**1.	Extension to solve blind video inverse problem on established video dataset.**

Our proposed method can be extended to address blind video inverse problems, such as video deblurring on the GoPro dataset [1]. Please refer to the revised Appendix B for the detailed explanations and experimental results.

**2.	Conduct experiments on additional video restoration method, dataset, and human perceptual evaluation.**

In response to the reviewers’ suggestions, we conducted experiments on an additional video restoration method (DiffIR2VR [2]), a high fps video dataset (Pexels [3]), and a human perceptual study involving computer vision researchers. The results demonstrate that our method consistently outperforms comparative methods across diverse datasets and evaluations. Please refer to the revised Appendix C.3-5 for further details.

**3.	Preview of future work on latent image diffusion models.**

In response to reviewer imTd’s suggestion, we included a preview of potential future work on latent image diffusion models. Our method is adaptable to SDXL [4], enabling broader applications and improved performance. Detailed discussions are available in the revised Appendix C.7.

For point-to-point response, please refer to below.

**References**

[1] Nah, Seungjun, et al. "Deep multi-scale convolutional neural network for dynamic scene deblurring." CVPR 2017.

[2] Yeh, Chang-Han, et al. "DiffIR2VR-Zero: Zero-Shot Video Restoration with Diffusion-based Image Restoration Models." arXiv preprint arXiv:2407.01519 (2024).

[3] Pexels.com, URL: https://www.pexels.com.

[4] Podell, Dustin, et al. “SDXL: Improving Latent Diffusion Models for High-Resolution Image Synthesis” ICLR 2024.

---

### Meta-Review · Area_Chair_5Txa · 2024-12-23

**Metareview:**

This paper addresses video inverse problems, or video restoration tasks, using a pretrained image-based diffusion model. Reviewers find the proposed extension to the spatio-temporal domain novel, and the experimental results confirm its effectiveness and efficiency. During the discussion stage, the authors addressed key concerns raised by the reviewers, including extending the approach to blind restoration and conducting more extensive evaluations on real-world datasets. The reviewers are largely satisfied with the authors’ responses and unanimously agree that this paper surpasses the acceptance threshold. The AC found no strong reasons to disagree and recommends accepting this paper at ICLR.

**Additional Comments On Reviewer Discussion:**

Although the authors demonstrated the feasibility of extending the method to blind restoration and evaluated its performance in real-world scenarios, a noticeable drop in performance was observed compared to synthetic cases. The AC strongly recommends that the authors investigate this issue further.

---

### Decision · Program_Chairs · 2025-01-22

Accept (Poster)